# Co-inhibition of ATM and ROCK synergistically improves cell proliferation in replicative senescence by activating FOXM1 and E2F1

Eun Jae Yang[1,9], Ji Hwan Park [1,9], Hyun-Ji Cho[1], Jeong-A Hwang[1], Seung-Hwa Woo[1], Chi Hyun Park[2], Sung Young Kim[3], Joon Tae Park[4], Sang Chul Park[5,6✉], Daehee Hwang [7✉] & Young-Sam Lee [1,5,8✉]

The multifaceted nature of senescent cell cycle arrest necessitates the targeting of multiple factors arresting or promoting the cell cycle. We report that co-inhibition of ATM and ROCK by KU-60019 and Y-27632, respectively, synergistically increases the proliferation of human diploid fibroblasts undergoing replicative senescence through activation of the transcription factors E2F1 and FOXM1. Time-course transcriptome analysis identified FOXM1 and E2F1 as crucial factors promoting proliferation. Co-inhibition of the kinases ATM and ROCK first promotes the G2/M transition via FOXM1 activation, leading to accumulation of cells undergoing the G1/S transition via E2F1 activation. The combination of both inhibitors increased this effect more significantly than either inhibitor alone, suggesting synergism. Our results demonstrate a FOXM1- and E2F1-mediated molecular pathway enhancing cell cycle progression in cells with proliferative potential under replicative senescence conditions, and treatment with the inhibitors can be tested for senomorphic effect in vivo.

[1] Department of New Biology, DGIST, Daegu 42988, Republic of Korea. [2] Department of Computer Science and Engineering, Kangwon National University, Chuncheon 24341, Republic of Korea. [3] Department of Biochemistry, Konkuk University School of Medicine, Seoul 05029, Korea. [4] Division of Life Sciences, College of Life Sciences and Bioengineering, Incheon National University, Incheon 22012, Republic of Korea. [5] Well Aging Research Center, Division of Biotechnology, DGIST, Daegu 42988, Republic of Korea. [6] The Future Life & Society Research Center, Advanced Institute of Aging Science, Chonnam National University, Gwangju 61469, Republic of Korea. [7] Department of Biological Sciences, Seoul National University, Seoul 08826, Republic of Korea. [8] New Biology Research Center, DGIST, Daegu 42988, Republic of Korea. [9] These authors contributed equally: Eun Jae Yang, Ji Hwan Park. ✉email: scpark@snu.ac.kr; daehee@snu.ac.kr; lee.youngsam@dgist.ac.kr

The progression of aging is characterized by the accumulation of senescent cells in living organisms[1,2]. Cells undergoing senescence exhibit a significant reduction in cell division, which is caused by suppression of the G1/S and G2/M transitions due to diverse cell cycle arrest mechanisms. For example, upregulation of p21 and p16 decreases the activity of the CDK2-cyclin E and CDK4/6-cyclin D complexes, respectively, resulting in the G1/S transition[3–5]. Additionally, p21 inhibits the G2/M transition by inactivating CDK1-cyclin B[6,7]. Many anti-aging agents, such as metformin and rapamycin, have been reported to inhibit senescence-associated secretory phenotypes (SASPs)[8,9], but their roles in ameliorating the senescence-associated decrease in cell proliferation remain unclear. Only a few agents that can modulate cell cycle-related factors have been reported to increase the proliferation of cells undergoing senescence. For example, epigallocatechin-3-O-gallate was reported to decrease the levels of p53 and p21 and increase the number of human diploid fibroblasts (HDFs) at S-phase in replicative senescence[10]. In addition, curcumin was reported to increase the division of human umbilical vein endothelial cells and decrease p21 expression in oxidative stress-induced senescence[11].

Cellular senescence is a multifaceted process elicited by diverse senescence-associated factors, such as DNA damage, loss of proteostasis, and impairment of subcellular organelles[12]. The inability of a single agent to target these different factors simultaneously is an intrinsic limitation imposed by the multifaceted nature of cellular senescence. Accordingly, the use of multiple agents, each targeting a different factor, has been suggested to collaboratively reinforce the reprogramming of the cell cycle network, thereby imparting synergism[13,14]. However, this combinatorial targeting approach has rarely been exploited to date. Previously, we identified two compounds, an ataxia-telangiectasia mutated (ATM) inhibitor (KU-60019; KU) and a rho-associated protein kinase (ROCK) inhibitor (Y-27632; Y). KU induced the functional recovery of lysosomes by modulating the phosphorylation of a V-type ATPase subunit (ATP6V1G1)[15], and Y promoted the functional recovery of the oxidative phosphorylation (OXPHOS) system in mitochondria by inhibiting the phosphorylation of mitochondrial Rac Family Small GTPase 1 (RAC1)[16]. However, the synergism of KU and Y, which have different modes of action, has not been investigated.

Here, we evaluated the combinatorial effect of KU and Y in replicative cellular senescence. Combination treatment led to synergism, as evidenced by the more effective suppression of senescence phenotypes, including senescence-associated cell cycle arrest, and impairment of mitochondrial or lysosomal functionality, even at a suboptimal concentration of Y. Time-course transcriptome analysis revealed two transcription factors (TFs)—FOXM1 and E2F1—as crucial regulators of the increase in cell proliferation. Spatiotemporal analysis of cell cycle regulators (CDKs and cyclins) further elucidated that the synergistic effect was mediated by promotion of the FOXM1-mediated G2/M transition resulting from activation of PLK1 and CDK2 via co-inhibition of ATM and ROCK, followed by accumulation of cells undergoing the E2F1-mediated G1/S transition. Therefore, our results suggest the existence of a FOXM1- and E2F1-mediated pathway enhancing cell cycle progression in cells with proliferative potential under replicative senescence conditions, and the anti-aging effects of this pathway can be tested in vivo.

## Results

### KU and Y synergistically restore the proliferation of senescent HDF cultures.
We sought to examine the synergistic effects of KU and Y, which inhibit the kinase activity of ATM (autophosphorylation of ATM) and ROCK (phosphorylation of MYPT1), respectively, on the proliferation of senescent HDF cultures (Supplementary Fig. 1). To this end, we first determined the optimal concentrations of KU and Y for cotreatment as 0.5 and 2.5 μM, respectively. These concentrations resulted in the maximum proliferation of high passage HDFs at 15 days post treatment (DPT) among the four tested concentrations of each drug (0.125, 0.25, 0.5, and 1.5 μM for KU; 2.5, 5, 7.5, and 10 μM for Y) (Fig. 1a). Compared with the single-drug treatments, cotreatment with KU and Y (KU + Y) at the corresponding optimal concentrations significantly increased ($P < 1.0 \times 10^{-3}$) cell proliferation even at 3 DPT, and the increase in proliferation became more evident at 8 and 15 DPT (Fig. 1b and Supplementary Fig. 2a, b). Finally, we examined the effects of KU, Y, and KU + Y on cell proliferation at 60, 90, and 150 DPT. After long-term exposure, the synergistic effect of KU + Y on cell proliferation was maintained up to 150 DPT and was comparable to that measured after short-term exposure (15 DPT) (Supplementary Fig. 2c, d). Taken together, these data suggest that inhibition of ATM and ROCK by KU + Y synergistically contributes to increasing the proliferation of senescent HDF cultures.

### KU and Y cooperatively suppress the senescence phenotypes of HDFs.
We next examined the synergistic effect of KU + Y on senescence-associated lysosomal impairment by measuring the amounts of β-gal and lipofuscin. SA-β-gal staining and flow cytometric analysis of lipofuscin revealed that cotreatment led to significantly ($P < 1.0 \times 10^{-3}$) greater decreases in β-gal and lipofuscin than the single-drug treatments at both 3 and 8 DPT (Fig. 1b and Supplementary Fig. 3a–c). To quantitatively assess the synergism, we calculated the coefficient of drug interaction (CDI), which indicates synergistic activation (>1) or inhibition (<1)[17], and evaluated the CDI values for cell proliferation and the amounts of β-gal and lipofuscin. The CDI profiles at 3, 8, and 15 DPT showed that synergism was evident even from an early post-treatment day (3 DPT) and was maintained at late post-treatment days (8 and 15 DPT) (Fig. 1c). Notably, unlike that of KU + Y versus Y, the synergistic effect of KU + Y versus KU alone on lysosomal function (as evaluated by the amounts of β-gal and lipofuscin) was weaker at 15 DPT than at 3 and 8 DPT, because lysosomal function is affected predominantly by KU, as mentioned above.

Moreover, we examined the long-term effects of KU, Y, and KU + Y on the above senescence phenotypes (SA-β-gal staining, lysosomal function, and cell size) at 60 and 90 DPT. Significant effects of KU, Y, and KU + Y on these phenotypes were observed even after long-term exposure. However, the differences in the effects of KU and KU + Y decreased slightly to varying degrees for each phenotype compared to those observed after short-term exposure (Supplementary Fig. 3d–g), suggesting that the synergistic effects of KU + Y on these senescence phenotypes were attenuated, probably due to the saturation effect of KU.

We also examined the effect of cotreatment on DNA damage and found that cotreatment led to greater reductions in the numbers of γH2AX and 53BP1 foci as well as in the length of comet tails originating from DNA double-strand breaks at 15 DPT than did either single-drug treatment (Supplementary Fig. 3h, i). To examine the long-term effect on DNA damage, we further performed neutral comet assays and counted γH2AX and 53BP1 foci at 60 DPT in high passage HDFs with KU, Y, or KU + Y. Even after long-term exposure (60 DPT), reduced DNA damage was still observed in the treated cells compared to the DMSO-treated control cells (Supplementary Fig. 3h, i). On the other hand, compared to short-term exposure, long-term exposure increased the absolute levels of DNA damage. This increased level of DNA damage in the long-term treated cells was consistent with the lack of a significant decrease in p16 and p21

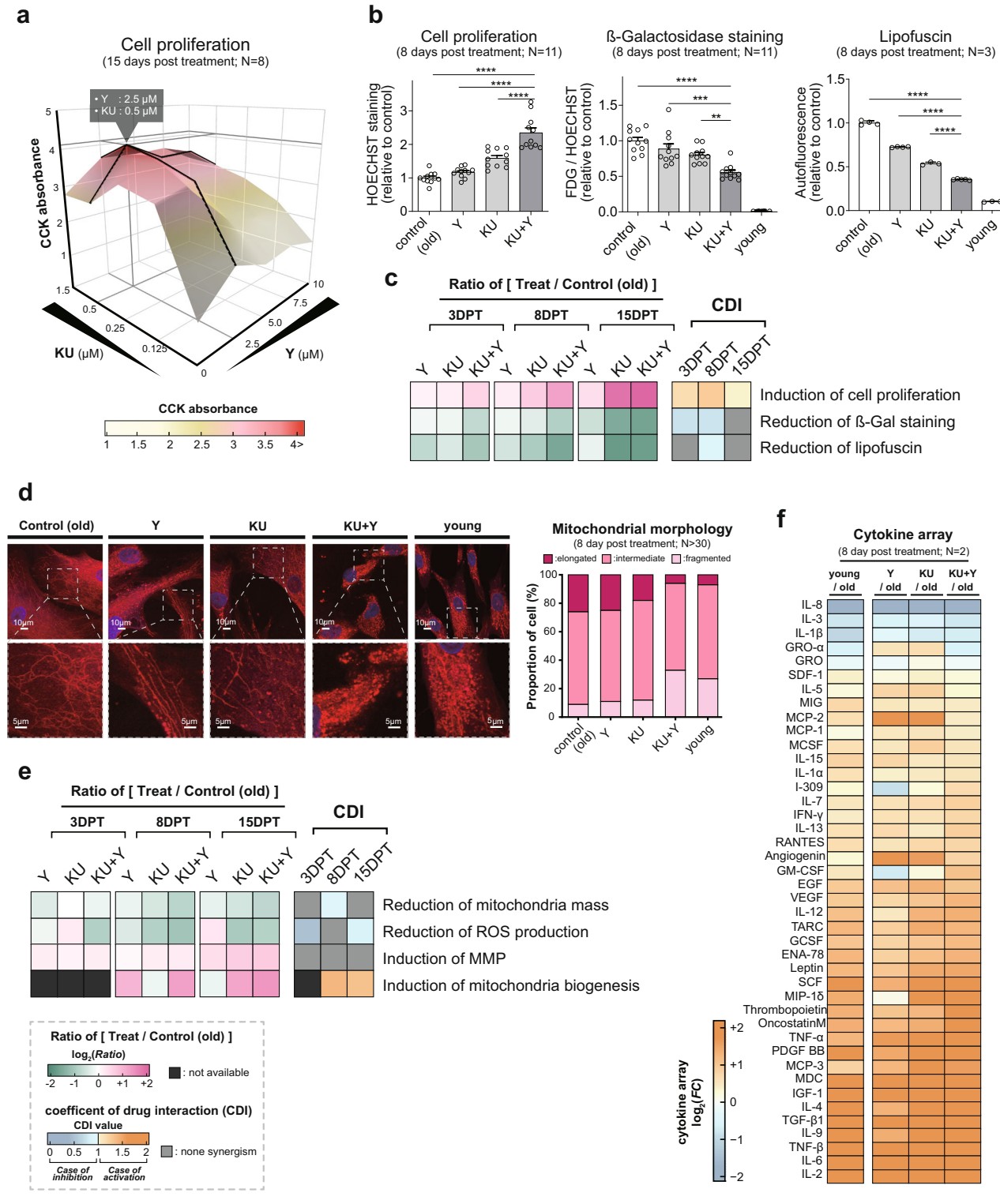

expression (Supplementary Fig. 3j). Of note, this expression pattern of p21 would be in line with the previously reported role of p21 in the induction of genomic instability[18].

Another common feature of KU and Y is their ability to improve mitochondrial function in senescent cells[15,16]. We thus hypothesized that the combined effect of KU and Y on the functionality of mitochondrial proteins may improve mitochondrial quality control. As expected, cotreatment led to stronger induction of mitochondrial fragmentation in high passage HDFs than did either single-drug treatment, restoring mitochondrial

fragmentation to a level similar to that observed in low passage HDFs (Fig. 1d). Additionally, several key features related to mitochondrial function were improved by cotreatment, as indicated by the decreases in total mitochondrial mass and ROS production, as well as the increases in mitochondrial biogenesis and mitochondrial membrane potential (MMP) (Supplementary Fig. 4a–d). The temporal CDI profiles confirmed the synergistic effects of KU and Y on the reductions in mitochondrial mass and ROS production and the induction of mitochondrial biogenesis from 3 to 15 DPT, but no synergism on MMP (Fig. 1e).

**Fig. 1 Synergistic effects of KU + Y on senescence phenotypes. a** Screening for the optimal concentrations of KU and Y based on their individual effects on cell proliferation (CCK absorbance). Senescent HDF cultures were treated with KU (0.125, 0.25, 0.5, and 1.5 μM) and Y (2.5, 5, 7.5, and 10 μM) for 15 days. Eight independent experiments were conducted per condition ($N = 8$). **b** Effects of Y, KU and KU + Y on cell proliferation ($N = 11$), SA-β-gal staining ($N = 11$), and lipofuscin staining ($N = 3$) at 8 DPT. The data are shown as the mean ± standard deviation (s.d.) values. **$P < 0.01$; ***$P < 1.0 \times 10^{-3}$; ****$P < 1.0 \times 10^{-4}$ by one-way ANOVA with Tukey's post hoc correction. **c** Log$_2$-fold changes in the phenotypic measurements shown in **b** between drug- and DMSO-treated senescent cells at 3, 8, and 15 DPT and CDIs calculated by the formula (efficacy of KU + Y)/(efficacy of KU) × (efficacy of Y) for the phenotypic measurements at each time point. **d** Representative confocal images showing the extent of elongation or fragmentation of mitochondria (red) at 8 DPT (top). A magnified (3×) image of the region enclosed in each dashed box in the corresponding top image is also shown (bottom). Nuclei were visualized by counterstaining with DAPI (blue). Proportions of cells containing elongated, intermediate, and fragmented mitochondria at 8 DPT with KU, Y, or KU + Y. In each cell, we calculated the percentage of elongated mitochondria >10 μm, and we classified a cell as containing elongated, intermediate, or fragmented mitochondria when this percentage was >70%, between 30 and 70%, or <30%, respectively. $N > 30$ cells per condition. **e** Log$_2$ fold changes in the measured values of the indicated mitochondrial phenotypes between drug- and DMSO-treated senescent cells at 3, 8, and 15 DPT and CDIs for the mitochondrial phenotypic measurements at each time point. **f** Heatmap showing the log$_2$ fold changes in cytokine abundances measured by the cytokine array assay for the indicated comparisons (e.g., young/old indicates young versus old). Up- and downregulation are represented by blue and orange, respectively. The color bar shows a gradient representation of the log$_2$ fold change values.

To examine the synergistic effect of KU + Y on SASPs, we next measured the levels of cytokines in senescent cells treated with KU, Y, or KU + Y at 8 DPT using an Abcam Human Cytokine Antibody Array, which contains antibodies against 42 cytokines, and compared these levels with those in DMSO-treated control cells. Of the 42 cytokines, 38 showed a trend toward upregulation and 4 showed a trend toward downregulation after treatment (Fig. 1f and Supplementary Data 1). KU, Y, and KU + Y appeared to restore these levels in high passage HDFs ('old') toward the levels in low passage HDFs ('young'). However, the differences in the log$_2$-fold-change values between KU and KU + Y and between Y and KU + Y were not significant, suggesting that the synergistic effects of KU + Y on SASPs might not be apparent. Taken together, these data suggest that inhibition of ATM and ROCK by KU + Y cooperatively contributes to the suppression of key senescence phenotypes (i.e., lysosomal and mitochondrial functional impairment, ROS generation, and DNA damage).

**Transcriptome analysis reveals key targets of the synergistic effect of KU + Y.** To examine the molecular nature of the synergistic effects of KU + Y on the aforementioned senescence phenotypes, we performed gene expression profiling of high passage HDFs treated with DMSO (control), KU, Y, or KU + Y at 3, 8, and 15 DPT, as well as low passage HDFs (Fig. 2a and Supplementary Data 2). Using the gene expression profiles, we first identified 4,904 senescence-associated genes (SAGs) as up- or downregulated in high passage HDFs compared to low passage HDFs (Fig. 2a, 'SAGs'). From these SAGs, we further selected 1432 drug-associated genes (DAGs) as those with restored expression patterns in high passage HDFs after treatment with KU (435 DAG$_{KU}$), Y (13 DAG$_Y$), or KU + Y (1404 DAG$_{KU+Y}$) (Fig. 2a, 'DAGs'; Supplementary Fig. 5a, b). Among the 1404 DAG$_{KU+Y}$, we finally selected 914 synergism-associated genes (SynAGs) as those up- or downregulated in KU + Y-treated high passage HDFs compared to KU- or Y-treated high passage HDFs at each time point (Fig. 2a, 'SynAGs'): 39 SynAGs at 3 DPT (SynAG03); 203 SynAGs at 8 DPT (SynAG08); and 779 SynAGs at 15 DPT (SynAG15). Of these SynAGs, 12 were shared across all time points, 69 were detected at both 8 and 15 DPT, and 691 were detected only at 15 DPT (Fig. 2b).

To systematically examine the temporal characteristics of the SynAGs, we next categorized them into 14 groups based on their patterns of up- and downregulation at the three different time points (Supplementary Fig. 5c). Among these groups, we focused on the following 7 major groups that contained >10 SynAGs (Fig. 2c): group (1) 10 genes upregulated at all three DPT (EML-UP); group (2) 68 genes upregulated at both 8 and 15 DPT (ML-UP); groups (3 and 4) 105 genes upregulated at 8 DPT (M-UP) and 377 genes

upregulated at 15 DPT (L-UP); and groups (5, 6, and 7) 11 genes downregulated at 3 DPT (E-DOWN), 10 genes downregulated at 8 DPT (M-DOWN), and 314 genes downregulated at 15 DPT (L-DOWN). To understand the cellular processes associated with these groups of SynAGs, we then carried out gene ontology (GO) biological process (GOBP) enrichment analysis of the genes in each major group (Supplementary Data 3). The groups of upregulated SynAGs (EML-, ML-, M-, and L-UP) were associated mostly with processes related to the cell cycle or cell division (Fig. 2d). These data collectively indicate that KU + Y leads to much more robust cell proliferation than KU or Y alone. On the other hand, the processes related to calcium ion homeostasis and protein transport (including extracellular vesicle and Golgi apparatus) were enriched with the downregulated SynAGs (E- and L-DOWN groups, respectively).

The presence of these SynAGs suggests the existence of key TFs activated by KU + Y and indicates that these TFs control the transcriptional regulation of the SynAGs. To identify the key TFs, we performed TF enrichment analysis of each major SynAG group and found that the TFs FOXM1 and E2F1 had a significant ($P < 0.05$) number of target genes in the SynAG groups (Fig. 2e and Supplementary Data 4). FOXM1 and E2F1 are essential regulators involved in the release of G1/S (E2F1[19,20]) and G2/M (FOXM1[21,22]) arrest, a critical feature of the synergistic enhancement of cell proliferation by KU + Y. Recently, a cell cycling-promoting function of FOXM1 was reported[23–26]. We next confirmed the upregulation of FOXM1 and E2F1 at 3, 8, and 15 DPT with KU, Y, or KU + Y using quantitative real-time polymerase chain reaction (qRT-PCR). The temporal expression patterns of FOXM1 and E2F1 showed faster induction of FOXM1 than of E2F1 by KU, Y, and KU + Y (Fig. 2f). Importantly, among the three treatments, KU + Y led to the fastest, strongest induction of FOXM1, indicating strong synergism.

Moreover, FOXM1 and E2F1 show patterns of gradually decreasing expression during human aging, particularly in highly proliferative tissues such as dermal skin, according to the transcriptome dataset of human dermal fibroblasts[27] (Fig. 2g), consistent with previous in vivo findings[23–25,28–30]. We further analyzed the senescence-associated changes in FOXM1 and E2F1 expression in Hutchinson-Gilford progeria syndrome (HGPS)[27] and in other cellular senescence models[31]. Consistent with the patterns in the normal aging dataset, FOXM1 and E2F1 were downregulated in HGPS skin fibroblasts as well as under senescence conditions compared to their levels in healthy controls (Fig. 2h and Supplementary Fig. 6). Taken together, these data suggest that FOXM1 and E2F1 can act as key TFs that control the transcriptional induction of the SynAGs required for cell cycle activation in senescent cell cultures.

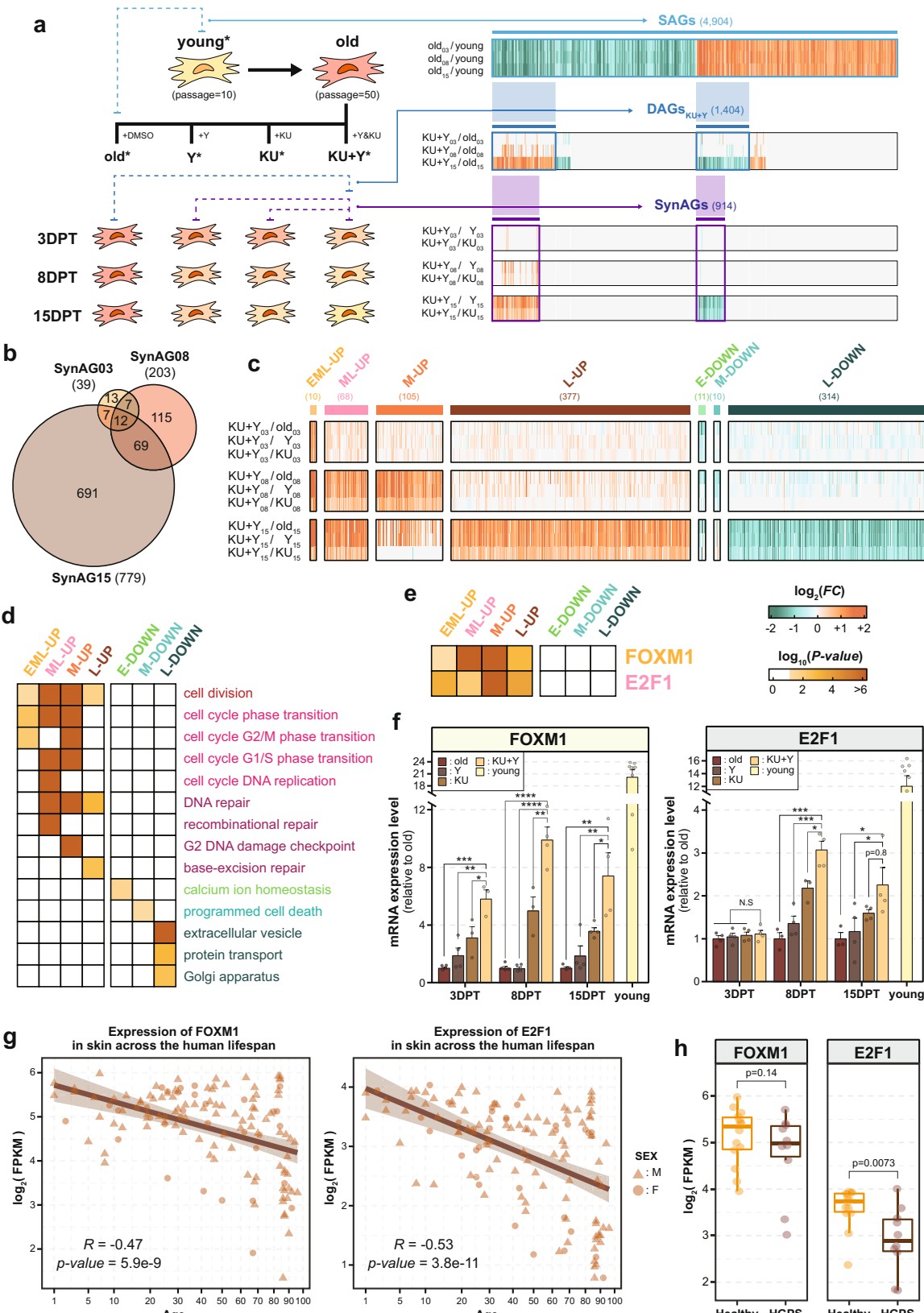

**FOXM1 is activated synergistically by KU and Y at the early stage after treatment.** To confirm whether FOXM1 is essential for the observed synergistic effect on cell proliferation, we knocked down *FOXM1* in high passage HDFs using shRNA #1–3 against *FOXM1* (Fig. 3a, bottom) and compared the proliferation of FOXM1 knockdown and nontargeting shRNA-transduced

HDFs treated with KU + Y at 8 DPT. FOXM1 knockdown abrogated the ability of KU + Y to restore cell proliferation (Fig. 3a, top). The same abrogation was observed consistently after treatment with thiostrepton, a FOXM1 inhibitor[32] (Supplementary Fig. 7a). We next overexpressed FOXM1 in high passage HDFs and compared the counts of HDFs overexpressing

**Fig. 2 Key candidate regulators of the synergistic effects of KU and Y on cell proliferation. a** Overall scheme for the identification of key regulator candidates mediating the synergistic effects of KU + Y on cell proliferation. See the main text for detailed descriptions. **b** Venn diagram showing the relationships among SynAGs identified from transcriptome data generated at 3 (SynAG03), 8 (SynAG08), and 15 (SynAG15) DPT. **c** Up- (orange) or downregulation (green) patterns of the SynAGs in the 7 major groups (EML-, ML-, M-, and L-UP; E-, M-, and L-DOWN) in the comparisons of senescent vs. young HDFs (old/young); KU + Y- vs. DMSO-treated high passage HDFs (KU + Y/old); and KU + Y-treated vs. either KU-treated (KU + Y/KU) or Y-treated (KU + Y/Y) high passage HDFs at 3, 8, or 15 DPT. The number in parentheses indicates the number of SynAGs in each major group. **d** Cellular processes enriched with the SynAGs in each of the seven major groups. **e** Enrichment of TF target genes in the seven major groups. The significance of enrichment (P-value) for each cellular process or TF enriched with the SynAGs in the major groups is indicated by the color. The color bars show a gradient representation of the $\log_{10}$(enrichment P-value). **f** Temporal changes in the mRNA expression levels of FOXM1 and E2F1, as measured by qRT-PCR, at 3, 8, and 15 DPT with DMSO, KU, Y, and KU + Y. The transcript levels of the TFs were first normalized to that of RPS11, and these normalized levels were further normalized to those in old (DMSO-treated) HDFs. Means ± SEMs; $N = 3$–4 per experiment. $*P < 0.05$; $***P < 1.0 \times 10^{-3}$; $****P < 1.0 \times 10^{-4}$ compared with KU + Y by one-tailed Student's t-test. **g** Aging-associated expression patterns of FOXM1 and E2F1 mRNA in human dermal fibroblasts. **h** Expression patterns of FOXM1 and E2F1 mRNA in HGPS skin fibroblasts.

FOXM1 and overexpressing mCherry (control) at 4, 9, 15, 22, 29 and 38 days after transduction. FOXM1 overexpression led to increased proliferation (Supplementary Fig. 7b). These data indicate that FOXM1 is a critical regulator that mediates the activating effect of KU + Y on the cell cycle.

To function as an active TF, FOXM1 must be translocated into the nucleus[33]. Nuclear localization of FOXM1 was observed beginning at 8 h post treatment with KU, Y, and KU + Y (Supplementary Fig. 8). However, nuclear localization of FOXM1 increased more predominantly over time after treatment with KU + Y than after treatment with KU or Y alone beginning at 3 DPT (Fig. 3b, c). The above data showed that nuclear translocation of FOXM1 preceded the proliferation of high passage HDFs upon KU + Y treatment, suggesting that FOXM1 can act as an early regulator to mediate cell cycle activation by KU + Y treatment. To test this hypothesis, we first examined whether eight representative SynAGs (TK1, CDK1, CDC20, CCNB1/2, BIRC5, KIF20A, and E2F1) involved in the cell cycle are transcriptionally regulated by FOXM1. Thiostrepton treatment led to the suppression of KU + Y-induced upregulation of these genes (Supplementary Fig. 9a).

We further performed chromatin immunoprecipitation (ChIP)-qPCR to examine whether FOXM1 binds to the promoters of five representative cell cycle-related SynAGs (CDK1, CCNB1, BIRC5, KIF20A, and E2F1) after KU + Y treatment. FOXM1 showed significantly stronger binding to the promoters of the five SynAGs in KU + Y-treated cells than in KU- or Y-treated cells (Fig. 3d and Supplementary Fig. 9b). Finally, the protein level of FOXM1 increased beginning at 2 DPT with KU + Y, and the synergistic effect on the FOXM1 protein level became apparent beginning at 8 DPT (Supplementary Fig. 10a). Taken together, these data indicate that FOXM1 is an early regulator of the synergistic effect of KU + Y on the cell cycle.

**E2F1 is synergistically activated by KU and Y at later stages after treatment**. E2F1 is an important TF that regulates genes needed for activating the G1/S transition during the cell cycle[34,35]. To determine whether E2F1 is involved in the synergistic effect of KU + Y on cell proliferation, we knocked down E2F1 in high passage HDFs using shRNAs #1–3 against E2F1 (Fig. 4a, bottom) and compared the proliferation of E2F1 knockdown and non-targeting shRNA-transduced HDFs treated with KU + Y at 8 DPT. Similar to FOXM1 knockdown, E2F1 knockdown abrogated the ability of KU + Y to restore cell proliferation (Fig. 4a, top). In addition, similar to FOXM1 overexpression, E2F1 overexpression delayed cellular proliferation arrest (Supplementary Fig. 7b). We next examined the nuclear localization of E2F1 at 3 and 8 DPT with KU + Y and found that the level of nuclear E2F1 increased beginning at 8 DPT, which was later than the increase

in nuclear FOXM1 (Fig. 4b, c). We finally investigated whether E2F1 actually binds to the promoters of four representative E2F1 target cell cycle-related SynAGs (BIRC5, PTTG1, UBE2C, and PRC1) in high passage HDFs treated with KU, Y, or KU + Y. E2F1 showed significantly stronger binding to the promoters of the representative SynAGs in KU + Y-treated cells than in KU- or Y-treated cells (Fig. 4d and Supplementary Fig. 9c). Moreover, the protein level of E2F1 increased beginning at 8 DPT with KU + Y (Supplementary Fig. 10b), consistent with the aforementioned timing of E2F1 nuclear localization. Notably, this time was much later than that (2 DPT) of FOXM1 nuclear localization. The synergistic effect on the E2F1 protein level became apparent beginning at 8 DPT. Taken together, these results indicate that E2F1 is a regulator of the synergistic effect of KU + Y on cell proliferation at a later stage after treatment.

**KU + Y activates the CDK1-cyclin A/B pathway at the early stage and the CDK2-cyclin D/E pathway at the late stage by suppressing p16/p21-mediated inhibition**. We found hyperactivation of the ATM and ROCK kinases in senescent HDF cultures (Supplementary Fig. 1), consistent with previous reports[15,16]. ATM inhibits CDK2 activation by activating CHK2, which inhibits the CDC25A-CDK2 pathway[15,36–38], whereas ROCK suppresses CDK1 activation by inhibiting the AKT-PLK1-CDC25C-CDK1 pathway[39–41]. The CDK1-cyclin A/B pathway is involved in the G2/M transition, while the CDK2-cyclin A/D/E pathway, which is inhibited by p16 and p21, is involved in the G1/S transition[42–44]. We next sought to determine how inhibition of ATM and ROCK by KU and Y, respectively, affects the phosphorylation or expression of these regulators. To this end, we measured changes in the phosphorylation of CHK2, TP53, AKT, PLK1, CDC25A/B/C, and CDK1/2 and the levels of CDK1/2, cyclins A/B/D/E, p16, and p21 in high passage HDFs at 1 h and at 1, 3, and 8 days after treatment with KU, Y, and KU + Y. KU reduced the phosphorylation of CHK2 (p-Thr68), while Y increased the phosphorylation of AKT (p-Ser473) at 1 h (Supplementary Fig. 11). Interestingly, KU + Y synergistically increased the levels of cyclins A and B beginning at 1 DPT and 3 DPT, respectively, consistent with the timing of FOXM1 activation, suggesting activation of the G2/M transition at the early stage after treatment (Fig. 5a–c). On the other hand, KU + Y did not lead to an effect on the p16 and p21 levels at the early stage (Fig. 5b) but decreased these levels beginning at 8 DPT (Fig. 5d, e); however, combination treatment synergistically increased both the abundance and phosphorylation of CDK2 and retinoblastoma (Rb) beginning at 1 DPT (Fig. 5a, b and Supplementary Fig. 12). These data suggest that the G1/S transition is ready for activation beginning at 1 DPT but is not actually activated until a later stage when the expression of the inhibitors p16 and p21 is suppressed. Moreover, our subsequent results supported this synergistic

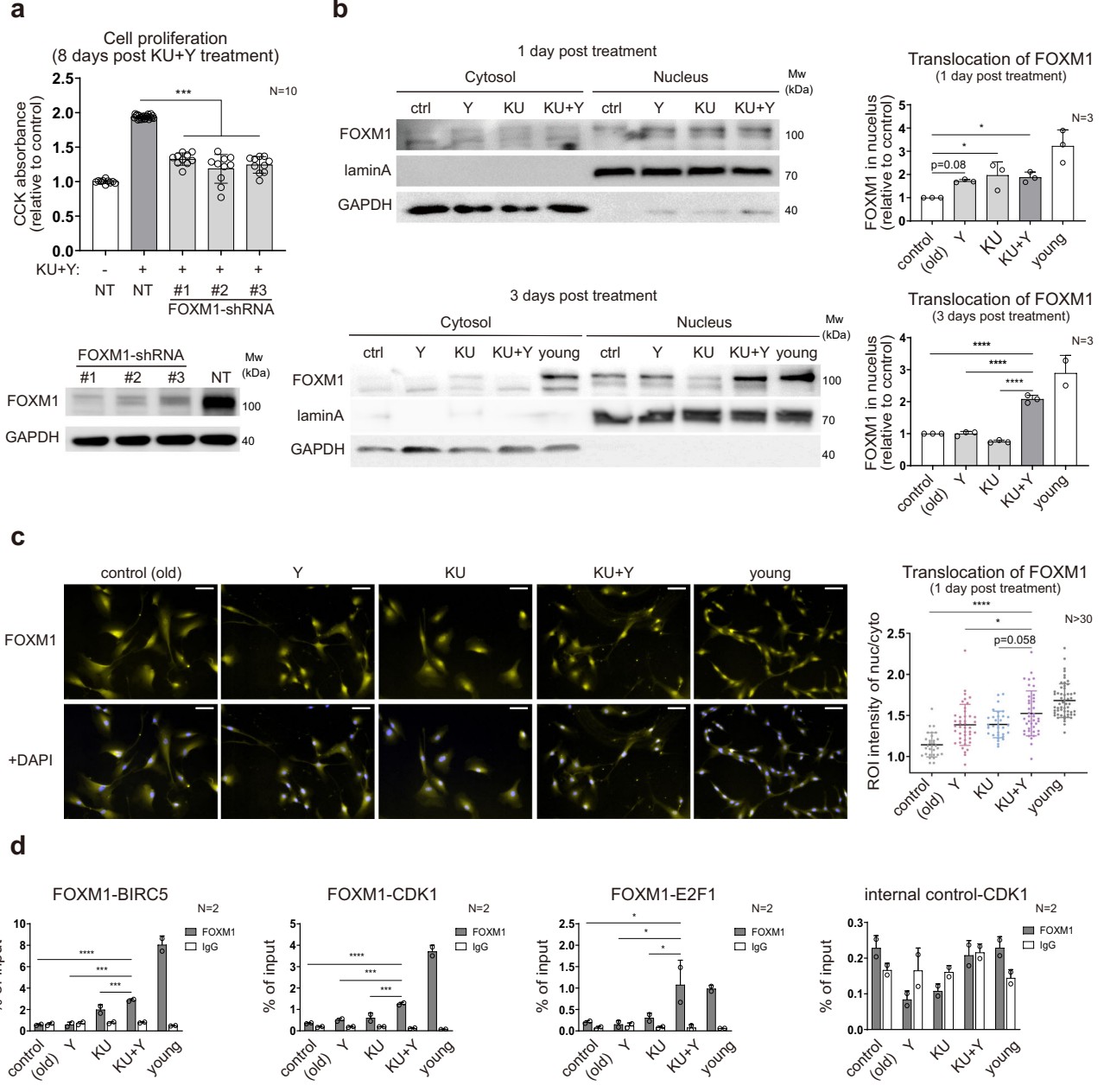

**Fig. 3 FOXM1 as a regulator of cell proliferation activated at the early stage after treatment. a** Attenuated proliferation of KU + Y-treated senescent HDF cultures by FOXM1 knockdown (top) and immunoblots of FOXM1 in senescent HDF cultures after FOXM1 knockdown (bottom). Cell proliferation was measured at 8 DPT. The data are shown as the mean ± s.d. values; $N = 10$ per condition. NT, nontargeting shRNA. **b** Immunoblots of the cytosolic and nuclear fractions probed with the anti-FOXM1 antibody after KU + Y treatment (1 and 3 DPT) and quantification of nuclear FOXM1. Lamin A and GAPDH were used as the loading controls for the nuclear and cytoplasmic fractions, respectively. The data are shown as the mean ± s.d. values; $N = 3$ per experiment. **c** Immunocytochemistry and quantification of nuclear FOXM1 in senescent cells after KU + Y treatment (1 DPT). Representative images showing the localization of FOXM1 (yellow; top and bottom) and costaining of FOXM1 and DAPI (blue, bottom) under each condition. The ratios of the nuclear FOXM1 intensity to the cytoplasmic FOXM1 intensity, defined as the DAPI-overlapping and non-DAPI-overlapping FOXM1 intensities, respectively, were quantified from the images. Scale bar = 50 μm. The data are shown as the mean ± s.d. values; $N > 30$ independent cells per sample. **d** Binding affinity of FOXM1 for the promoters of the indicated target genes, as measured by ChIP-qPCR. The data are shown as the mean ± s.d. values; $N = 2$ per experiment. For statistical analyses, **a–c** *$P < 0.05$; ***$P < 1.0 \times 10^{-3}$; ****$P < 1.0 \times 10^{-4}$ by one-way ANOVA with Tukey's post hoc correction. **d** *$P < 0.05$; ***$P < 1.0 \times 10^{-3}$; ****$P < 1.0 \times 10^{-4}$ by two-way ANOVA with Tukey's post hoc correction.

activation: (1) PLK1 activation mediated the phosphorylation of CDC25C (p-Ser198) and CHK2 inhibition mediated the dephosphorylation of CDC25A (at Ser124) and CDC25C (at Ser216) at 1 DPT (Supplementary Fig. 13), and (2) the phosphorylation of CDK1/2 (at Thr161 of CDK1 and at Thr160 of CDK2) increased, (Supplementary Fig. 12); in addition, the level of cyclin A

increased beginning 1 DPT, and these increase became more evident at 3 DPT (Fig. 5a–c).

According to previous reports, ATM also regulates CDK1 activation by inhibiting PLK1, and ROCK regulates CDK2 activation via AKT[45–52], suggesting that CDK1/2 may be synergistically activated by KU and Y via PLK1, a central

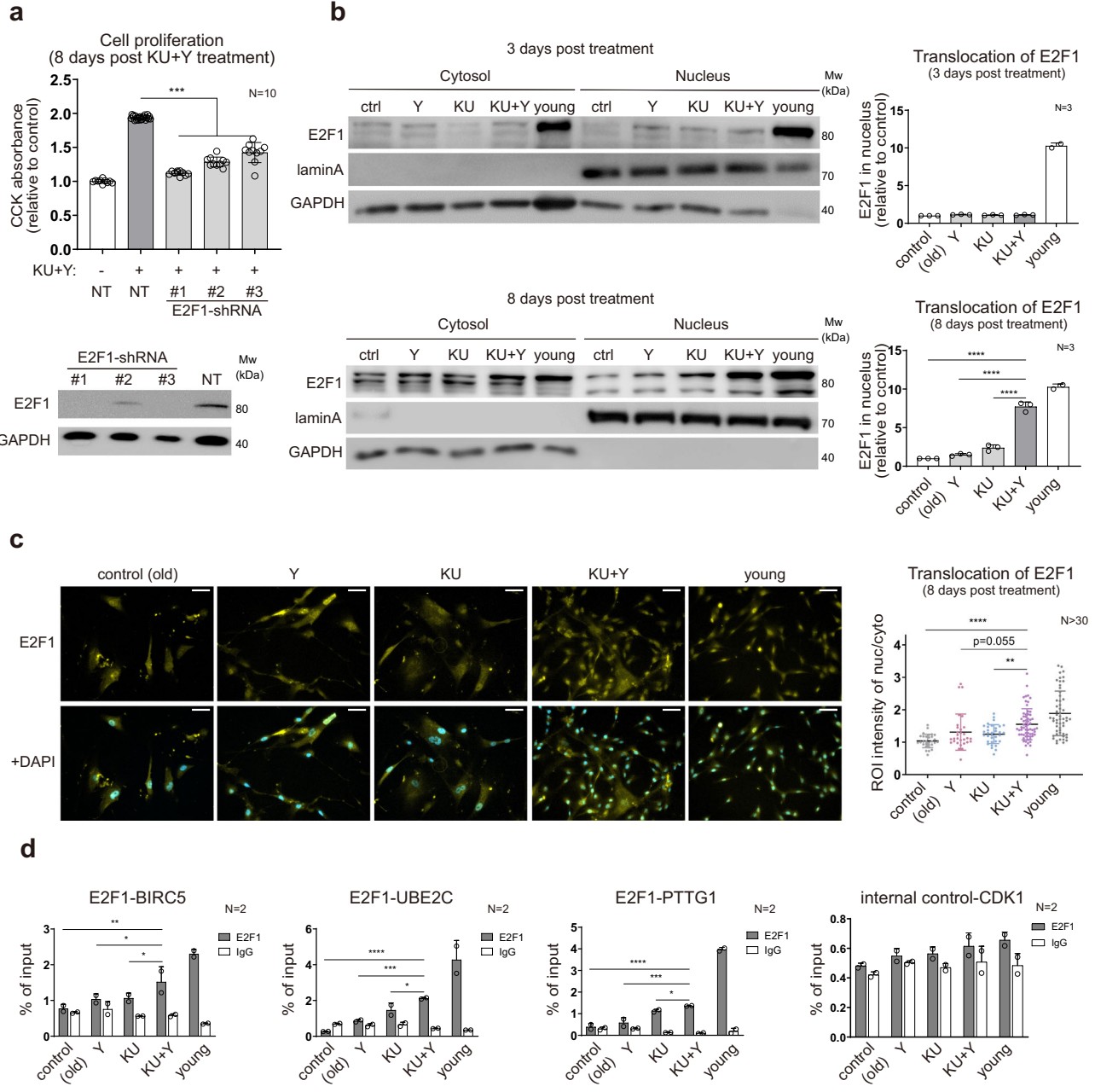

**Fig. 4 E2F1 as a key regulator of cell proliferation at the late stage after treatment. a** Attenuated proliferation of KU + Y-treated senescent HDF cultures by E2F1 knockdown (top) and immunoblots of E2F1 in senescent HDF cultures after E2F1 knockdown (bottom). Cell proliferation was measured at 8 DPT. The data are shown as the mean ± s.d. values; N = 10 per condition. NT, nontargeting shRNA. **b** Immunoblots of the cytosolic and nuclear fractions probed with the anti-E2F1 antibody after KU + Y treatment (3 and 8 DPT) and quantification of nuclear E2F1. Lamin A and GAPDH were used as the loading controls for the nuclear and cytoplasmic fractions, respectively. The data are shown as the mean ± s.d. values; N = 3 per experiment.
**c** Immunocytochemistry and quantification of nuclear E2F1 in senescent cells after KU + Y treatment (8 DPT). Representative images show the localization of E2F1 (yellow; top and bottom) and costaining of E2F1 and DAPI (blue, bottom) under each condition. The ratios of the nuclear E2F1 intensity to the cytoplasmic E2F1 intensity, defined as the DAPI-overlapping and non-DAPI-overlapping E2F1 intensities, respectively, were quantified from the images. Scale bar = 50 μm. The data are shown as the mean ± s.d. values; N > 30 independent cells per sample. **d** Binding affinity of E2F1 for the promoters of the indicated target genes, as measured by ChIP-qPCR. The data are shown as the mean ± s.d. values; N = 2 per experiment. For statistical analyses,
**a**–**c** **$^{**}P < 0.01$; $^{***}P < 1.0 \times 10^{-3}$; $^{****}P < 1.0 \times 10^{-4}$ by one-way ANOVA with Tukey's post hoc correction. **d** $^{*}P < 0.05$; $^{**}P < 0.01$; $^{***}P < 1.0 \times 10^{-3}$; $^{****}P < 1.0 \times 10^{-4}$ by two-way ANOVA with Tukey's post hoc correction.

regulator of the G2/M transition. To determine whether PLK1 can mediate the synergistic effect of KU + Y on cell proliferation, we inhibited PLK1 using volasertib and BI-2536 and measured cell proliferation at 8 DPT. As expected, inhibition of PLK1 led to significant reductions in the effects of both KU and Y

(Supplementary Fig. 14a). PLK1 inhibition prevented the translocation of FOXM1 even more strongly after KU + Y treatment (Supplementary Fig. 14b), thereby inhibiting the TF activity of FOXM1. Furthermore, we also knocked down (1) the V-type ATPase subunit ATP6V1G1, a target of ATM regulated by

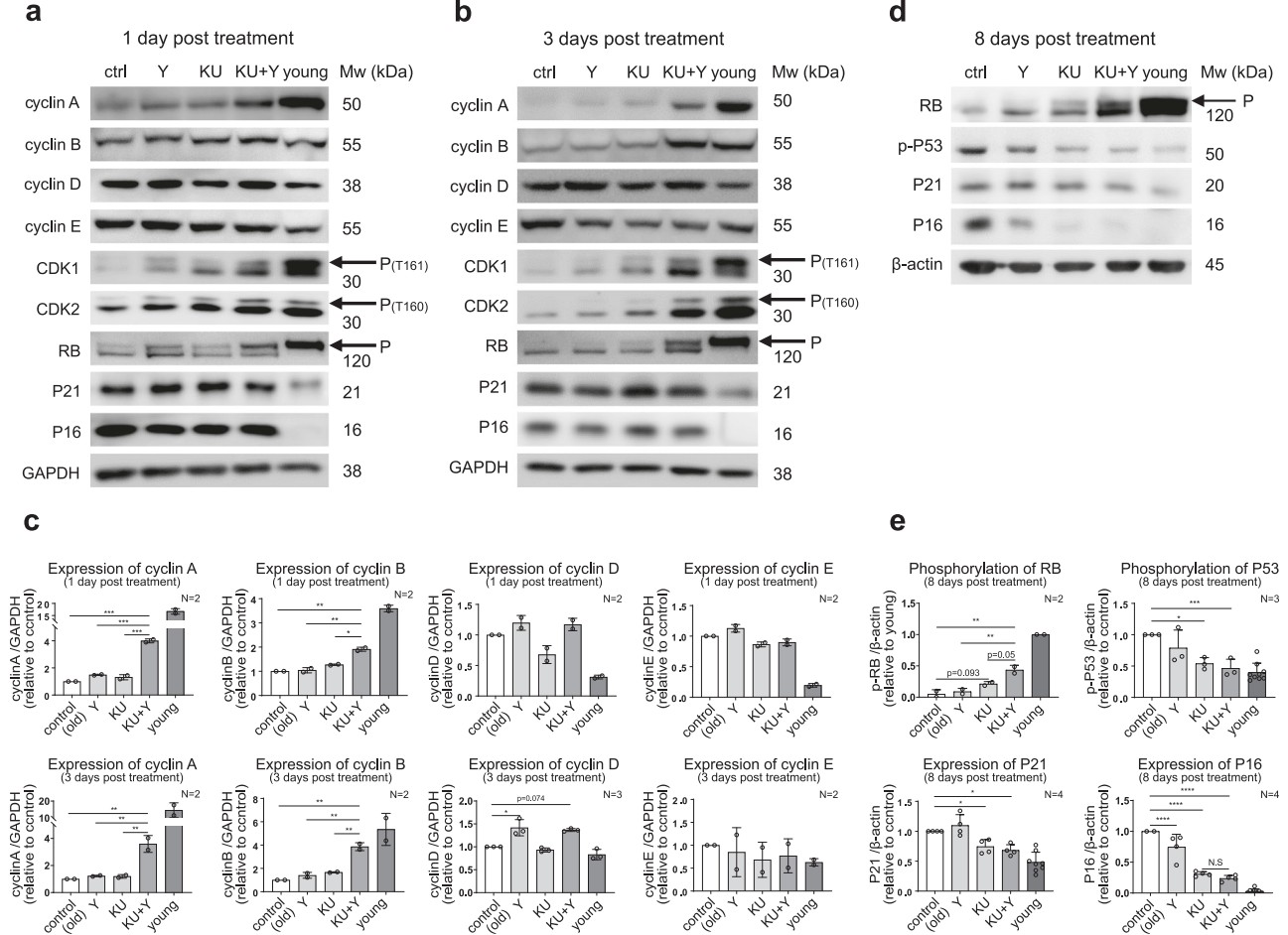

**Fig. 5 Induction of CDK-cyclin pathway activation by KU + Y at the early and late stages after treatment. a, b** and **d** Representative immunoblots of the indicated cell cycle regulators at 1 DPT (**a**), 3 DPT (**b**) or 8 DPT (**d**) with KU, Y, or KU + Y. The arrows indicate the bands representing the phosphorylated forms of the proteins. **c** and **e** The protein levels of cyclins (**c**), P21, and P16 (**e** bottom) and the levels of phosphorylated RB and P53 (**e** top) were quantified using images acquired from $N = 2$–4 samples per experiment. The data are shown as the mean ± s.d. values. $*P < 0.05$; $**P < 0.01$; $***P < 1.0 \times 10^{-3}$; $****P < 1.0 \times 10^{-4}$ by one-way ANOVA with Tukey's post hoc correction.

KU[15], and (2) RAC1B, a target of ROCK regulated by Y[16], and monitored the alteration of KU + Y-induced nuclear translocation of FOXM1 and E2F1 by the knockdown at 8 DPT. Interestingly, ATP6V1G1 knockdown did not affect the KU + Y-induced translocation of FOXM1 or E2F1, but RAC1B knockdown significantly inhibited the translocation of FOXM1 and E2F1 as well as the expression of E2F1 (Supplementary Fig. 14c). Consistently, the replicative senescent cultures of ATP6V1G1 knockdowned cells recovered cell proliferation by KU + Y treatment, but that of RAC1B knockdowned did not (Supplementary Fig. 14d). Previous our report showed that ATM-p53 axis is also involved in senescence amelioration[36]. In addition, RAC1B is known to promote cell proliferation and G1/S progression[53]. These data support the hypothesis that PLK1, as a potential crosstalk node of KU and Y, can mediate the synergistic effects of KU + Y on cell proliferation.

**KU + Y enhances the G2/M transition at the early stage followed by the G1/S transition at the late stage.** The above results suggest that KU + Y activates the FOXM1-mediated G2/M transition at the early stage after treatment and then activates the E2F1-mediated G1/S transition at the late stage. To test this hypothesis, we first performed cell cycle analysis of high passage HDFs at 1 and 8 DPT with KU, Y, or KU + Y. At the early stage (1 DPT), the proportion of G2/M-phase cells was decreased by

~9%, with no significant change in the proportion of G0/G1-phase cells (Fig. 6a). In contrast, the proportion of G0/G1-phase cells was reduced at the late stage (8 DPT), with no significant change in the proportion of G2/M-phase cells (Fig. 6b). We further sorted high passage HDFs into two groups of cells in the G1 and G2/M phases (Fig. 6c) and measured the proliferation of the cells in each group at 3 and 8 DPT. The cells in the G2/M group showed a much greater increase in proliferation at 3 DPT than those in the G1 group (Fig. 6d). In contrast, the cells in both the G1 and G2/M groups showed a far greater increase in proliferation at 8 DPT than at 3 DPT, with a similar degree of increase (Fig. 6e). These data collectively suggest that KU + Y first activates FOXM1, which promotes the G2/M transition in senescent HDF cultures with proliferative potential (a sub-population in G2 phase) at the early stage, and the cells accumulated after the G2/M transition can then undergo the G1/S transition via E2F1 activation at the late stage when p16 and p21 expression is suppressed.

## Discussion

Our results propose a mechanistic model that shows the dynamic regulation of cell cycle pathways by KU + Y inhibitors, leading to a synergistic effect on the proliferation of high passage cells undergoing replicative senescence. First, the model delineates the effects of the individual drugs: KU activates the CDK2-Rb

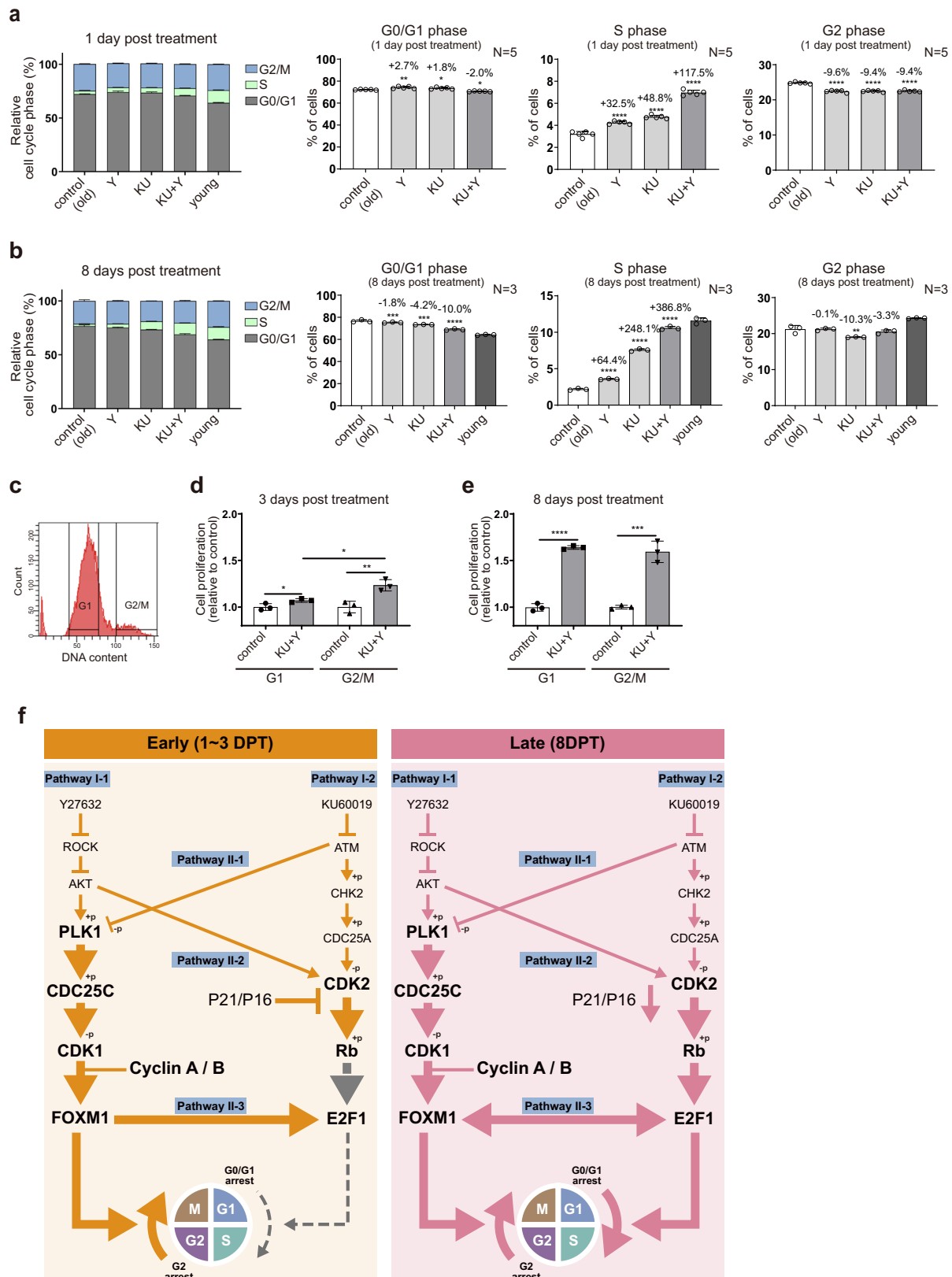

pathway via the ATM-CHK2-CDC25A pathway, and Y activates the CDK1/2-cyclin A pathway via the ROCK-AKT pathway (Fig. 6f, Pathways I-1/2). Second, the model further shows the crosstalk between ATM and ROCK in CDK1/2 activation (CDK1-cyclin A/B pathway activation via the ATM-PLK1-CDC25C pathway and CDK2 activation by AKT) and possibly

also the transcriptional regulation of E2F1 by FOXM1, which can explain the synergistic effect of KU + Y on the proliferation of high passage HDF cultures (Fig. 6f, Pathways II-1/2/3). Finally, the model describes the dynamic regulation of the G2/M and G1/S transitions: KU + Y first promotes the FOXM1-mediated G2/M transition in a subpopulation of cells with proliferative potential

**Fig. 6 Network model of the G2/M transition at the early stage and the G1/S transition at the late stage after treatment. a, b** Relative proportions of G0/G1-, S-, and G2/M-phase cells, as determined by cell cycle analysis with PI staining, at 1 DPT (**a**) or 8 DPT (**b**) with KU, Y, or KU + Y. The data are shown as the mean ± s.d. values; $N = 3$–5 per experiment. *$P < 0.05$; **$P < 0.01$; ***$P < 1.0 \times 10^{-3}$; ****$P < 1.0 \times 10^{-4}$ by one-way ANOVA with Tukey's post hoc test. **c** Distribution of DNA contents in high passage HDFs. The ranges of DNA contents used for sorting the cells in G1 and G2/M phases are indicated. **d, e** Cell proliferation in each sorted group at 3 (**d**) and 8 DPT (**e**). The data are shown as the mean ± s.d. values; $N = 3$ per experiment. *$P < 0.05$; **$P < 0.01$; ***$P < 1.0 \times 10^{-3}$; ****$P < 1.0 \times 10^{-4}$ by one-way ANOVA with Tukey's post hoc test. **f** Network model of the FOXM1-dependent G2/M transition at the early stage (1–3 DPT) after KU + Y treatment followed by the E2F1-dependent G1/S transition at the late stage (from 8 DPT). The arrows and suppression symbols indicate activation and inhibition reactions, respectively. The thick lines and large fonts (e.g., PLK1 at the early stage) indicate that the corresponding reactions and regulators, respectively, are activated at the indicated stage. Pathways I-1 and 2 are pathways affected by KU and Y, respectively, and pathways II-1, 2, and 3 are pathways of crosstalk between KU and Y. Finally, the thick arrow for cell cycle progression indicates an increase in the G2/M or G1/S transition.

at the early stage (Fig. 6f, left), and the cells accumulated after the G2/M transition then undergo the E2F1-mediated G1/S transition at the late stage when p16 and p21 expression is suppressed (Fig. 6f, right).

In this study, we elucidated a potential fundamental regulatory mechanism underlying the increased proliferation of cells undergoing replicative senescence. Previously, several molecular targets, including SIRT1, AMPK, and mTOR, associated with "calorie restriction-linked longevity pathways" for regulating cellular energy and metabolism have been identified, and modulation of their functions using activating (e.g., resveratrol for SIRT1 and metformin for AMPK) and inhibitory (e.g., rapamycin for mTOR) drugs has been demonstrated to ameliorate metabolic impairment and further extend the lifespan of organisms[54]. In addition, another type of agent that suppresses SASP production and senescence-associated secretion has been discovered (e.g., apigenin and kaempferol, which act by targeting the NF-κB pathway)[55]. Moreover, combination treatments with these drugs, similar to KU + Y in this study, have recently been introduced to synergistically suppress senescence phenotypes[13,14,56–58]. However, how these drugs modulate the cell cycle remains poorly investigated. Our findings suggest that the activated FOXM1-E2F1 pathway can constitute a fundamental regulatory axis for enhancing the proliferation of cells undergoing senescence.

FOXM1 and E2F1 undergo nuclear translocation to transcriptionally control diverse cell cycle-related genes upon treatment with KU and Y. In both U2OS cells and fibroblasts, phosphorylation of FOXM1 via the Raf/MEK/MAPK signaling pathway was shown to induce its nuclear translocation[59]. Moreover, nuclear import of E2F1 is promoted via the nucleoporin POM121-importin β axis[60]. Thus, the functional link of ATM/ROCK and MAPK signaling to the nuclear pore complex should be studied to understand the molecular mechanism underlying KU + Y-induced nuclear translocation of FOXM1 and E2F1. In addition to playing a role as an active TF in the nucleus, FOXM1 was recently demonstrated to translocate into mitochondria and regulate mitochondrial homeostasis independent of its nuclear function[61]. Because our results indicated that both KU and Y corrected mitochondrial dysfunction in senescent cells, whether mitochondrial translocation of FOXM1 can also be induced by KU + Y treatment—and, if so, whether mitochondrial FOXM1 is also linked to the recovery of mitochondrial function induced by KU + Y—awaits discovery.

FOXM1 and E2F1 are also upregulated in various types of cancers, such as hepatocellular carcinoma, myeloma, colon cancer, breast cancer, and lung cancer[62–69]. The use of KU + Y can thus raise concern because this combination may induce malignant transformation as a side effect. Interestingly, however, the oncogenes (BMP4, ADAMTS1, PLOD1, P4HA1, VEGFA, LOX, LOXL2, and MMP2) previously reported to be regulated by E2F1 or FOXM1 were not noticeably upregulated in high passage HDFs after KU + Y treatment. Also, in vivo cyclic induction of

FOXM1 has been just recently disclosed as a safe strategy to extend healthspan in progeria and natural aging mouse models[26]. Moreover, two independent studies[70,71] demonstrated that E2F1 transcriptionally regulates the expression of FOXM1 in MCF7 breast cancer cells, opposite the direction of transcriptional regulation shown in our study. These observations suggest that transcriptional regulation of FOXM1 by E2F1 might be more prevalent in cancer cells, while transcriptional regulation of E2F1 by FOXM1 is prevalent in senescent cells. Malignant transformation involves the occurrence of the E2F1-dependent G1/S transition before the FOXM1-dependent G2/M transition, while the increase in the proliferation of cells undergoing senescence first involves the FOXM1-dependent G2/M transition and subsequently the E2F1-dependent G1/S transition. These data collectively suggest that KU + Y may not significantly produce the side effect of malignant transformation, although this hypothesis should be thoroughly tested in detailed functional studies.

Herein, we addressed the possibility of FOXM1-E2F1 activation as a fundamental molecular mechanism for enhancing the proliferation of cells undergoing senescence. FOXM1-E2F1 activation not only modulates mitotic fidelity and SASP but also improves the proliferative capacity of high passage cells rather than clearing senescent cells, effectively managing not only the detrimental but also the beneficial (wound healing and tumor suppression) effects of senescence. Thus, the FOXM1-E2F1 pathway constitutes a functional axis with potential clinical benefit for extending the healthy lifespan. Accordingly, early upregulation of FOXM1 and subsequent induction of E2F1 after drug treatment can be used as a molecular gauge during drug screening or for optimization of combination treatment options (combination scheme, dose, treatment duration, etc.).

## Materials and methods

**Cell culture.** HDFs originating from neonatal skin (PCS-201-010; ATCC) were used in this study. Cells were cultured in Dulbecco's modified Eagle's medium [containing 25 mM glucose and supplemented with 10% fetal bovine serum (FBS; S001-01; Welgene), 100 units/ml penicillin, 10 μg/ml streptomycin, and 25 ng/ml amphotericin B (LS203-01; Welgene)] and maintained at 37 °C in 5% $CO_2$. Confluent cells were serially passaged at a 1:4 split ratio during low passages (up to passage 45) and a 1:3 (passages 46 and 47) or 1:2 split ratio during high passages (passages 48–50). Senescent HDF cultures were defined as populations with the following properties: >90% of the cells were positive for SA-β-gal staining, and the population doubling time (DT) of the cells was >14 days. HDFs with a DT of <1 day were defined as young cells. Cells were tested for mycoplasma contamination with a MycoAlert Mycoplasma Detection Kit (LT07-318; Lonza).

**Cell proliferation assay.** HDFs were plated into 96-well plates 3 days before the experiment. After drug treatment for 3, 8, or 15 days, 10 μl of D-Plus™ CCK reagent (Dongin) was added to each well, and the plates were placed in a cell culture incubator for 4 h. A plate reader (Tecan Infinite M200 PRO) was used to measure the absorbance at 450 nm. For the Hoechst 33342 staining assay, cells were stained with 10 μg/ml Hoechst 33342 for 30 min and washed twice with phosphate-buffered saline (PBS). The cell number was determined by measuring the fluorescence intensity using a Tecan Infinite M200 PRO plate reader at an excitation wavelength of 355 nm and an emission wavelength of 460 nm.

**SA-β-gal staining**. SA-β-gal assays were carried out with either a quantitative fluorescein di-β-d-galactopyranoside (FDG; Thermo Fisher Scientific) or an X-gal-based Senescence β-Galactosidase Staining Kit (Cell Signaling Technology, #9860). Cells were plated into 12-well (for the FDG assay) or 6-well (for the X-gal staining assay) plates 3 days before the experiment. The primary stock solution of FDG (200 mM) was prepared by dissolving FDG (5 mg) in 38 μl of a mixed solution of H₂O:DMSO (1:1). The secondary stock solution of FDG (2 mM) was prepared by diluting the primary stock solution with sterile H₂O at a 1:100 ratio. For the SA-β-gal assay, cells were fixed and stained with 1× β-galactosidase staining solution containing FDG (final concentration: 100 μM) or X-gal (final concentration: 1 mg/ml) following the manufacturer's protocol. The activity of SA-β-gal using FDG was determined by measuring the fluorescence intensity at excitation/emission wavelengths of 355 nm/460 nm. Each fluorescence intensity value was normalized to the cell number determined by Hoechst 33342 staining as mentioned above. Areas of cells positive for X-gal staining were randomly selected and visualized with an Olympus CKX41 microscope.

**Quantification of lipofuscin**. After two washes with PBS, cells were trypsinized, harvested in 500 μl of PBS, and injected into an LSR Fortessa (Beckton Dickson) flow cytometer. Autofluorescence from lipofuscin was measured at an excitation wavelength of 488 nm with a 530/30 nm bandpass filter (FITC channel).

**Immunocytochemistry**. Cells were seeded in 4-well chamber cell culture slides (SPL). After drug treatment for 3–15 days, the cells were fixed with 4% paraformaldehyde for 10 min and were then washed three times with PBS. Primary antibodies were diluted with gelatin dilution buffer (GDB) and added to the fixed cells for incubation at 4 °C overnight. After washing with PBS, secondary antibodies diluted 1:1000 with GDB were added for incubation for 1 h. Mounting solution with DAPI (Vector Laboratories) was used for counterstaining. Stained foci were detected with a Zeiss Axiocam camera. The following primary antibodies were used: rabbit anti-γH2AX (9218 S; 1:500 dilution; Cell Signaling Technology), mouse anti-53BP1 (05–726; 1:500 dilution; MilliporeSigma), mouse anti-FOXM1 (sc-376471; 1:250 dilution; Santa Cruz Biotechnology), and mouse anti-E2F1 (sc-193; 1:1000 dilution; Santa Cruz Biotechnology).

**Neutral comet assay**. Neutral comet assays were conducted with a Single Cell Gel Electrophoresis Assay kit (4250-050-K; Trevigen) according to the manufacturer's protocols with minor modifications. A total of $5 \times 10^5$ cells were resuspended in 0.5 ml of ice-cold PBS. The cell suspension (10 μl) was blended with 100 μl of warmed LMAgarose (4250-050-02; R&D Systems), and the mixture (50 μl) was then swiftly spread on slides as evenly as possible. DNA was stained with SYBR Gold (S-11494; Life Technologies), and comet tail lengths in 25–50 cells per condition were automatically estimated from the images using OpenComet V1.3 software in the image processing program ImageJ.

**Subcellular fractionation**. Cells were washed twice with PBS and harvested using a scraper. Harvested cells were lysed with detergent-free sucrose lysis buffer [10 mM HEPES (pH 7.4), 10 mM KCl, 1.5 mM MgCl₂, 300 mM sucrose, 0.5 mM DTT, 0.1% NP-40, and 0.5 mM PMSF]. After incubation on ice for 5 min, the lysates were centrifuged at 10,000 × g for 1 min at 4 °C, and the supernatants were prepared as the cytoplasmic fractions. The pellet fractions from the samples were further diluted with detergent-free glycerol lysis buffer [20 mM HEPES (pH 7.4), 100 mM KCl, 100 mM NaCl, 0.2 mM EDTA, 20% glycerol, 0.5 mM DTT, and 0.5 mM PMSF], incubated on ice for 1 h, and centrifuged at 10,000 × g for 5 min at 4 °C. The resulting supernatants were considered the nuclear fractions.

**Western blot analysis**. Cells were lysed in 2× Laemmli sample buffer (#161–0737, Bio–Rad) containing 5% β-mercaptoethanol (Sigma) and boiled at 95 °C for 5 min. Proteins in the lysates were gradually separated by sodium dodecyl sulfate–polyacrylamide gel electrophoresis (SDS–PAGE) and transferred onto polyvinylidene difluoride (PVDF) membranes (Millipore). The membranes were blocked with 5% bovine serum albumin (BSA; Sigma) diluted in Tris-buffered saline with 0.05% Tween 20 and incubated with primary antibodies. Next, the membranes were incubated with horseradish peroxidase (HRP)-conjugated secondary antibodies. Specific proteins were detected with enhanced chemiluminescence solution (P90720; Millipore) in an ImageQuant LAS-4000 digital imaging system (GE Healthcare). The primary antibodies used for western blotting were as follows: mouse anti-FOXM1 (sc-376471; 1:250 dilution; Santa Cruz Biotechnology), mouse anti-E2F1 (sc-193; 1:1000 dilution; Santa Cruz Biotechnology), rabbit anti-ATM (2873 S; 1:1000 dilution; Cell Signaling Technology), rabbit anti-phospho-ATM (phospho-Ser1981; 4526 S; 1:1000 dilution; Cell Signaling Technology), rabbit anti-MYPT1 (2634 S; 1:1000 dilution; Cell Signaling Technology), rabbit anti-phospho-MYPT1 (phospho-Thr835; 4563 S; 1:1000 dilution; Cell Signaling Technology), rabbit anti-Chk2 (05–649; 1:500 dilution; Millipore), rabbit anti-phospho-Chk2 (phospho-Thr68; 2661; 1:500 dilution; Cell Signaling Technology), rabbit anti-phospho-p53 (phospho-Ser15; 9284; 1:500 dilution; Cell Signaling Technology), rabbit anti-Akt (9272; 1:1000 dilution; Cell Signaling Technology), mouse anti-phospho-Akt (phospho-Ser473; 05–1003; 1:1000 dilution; Millipore), mouse anti-Rb (554136; 1:500 dilution; BD Bioscience), rabbit anti-

phospho-PLK1 (phospho-Thr210; 5472 T; 1:1000 dilution; Cell Signaling Technology), rabbit anti-PLK1 (4513 T; 1:1000 dilution; Cell Signaling Technology), rabbit anti-phospho-CDC25A (phospho-Ser124; ab156574; 1:1000 dilution; abcam), rabbit anti-CDC25A (3652 S; 1:1000 dilution; Cell Signaling Technology), rabbit anti-phospho-CDC25B (phospho-Ser353; orb571847; 1:1000 dilution; Biorbyt), rabbit anti-CDC25B (9525; 1:1000 dilution; Cell Signaling Technology), rabbit anti-phospho-CDC25C (phospho-Ser216; 4901 T; 1:1000 dilution; Cell Signaling Technology), rabbit anti-phospho-CDC25C (phospho-Ser198; 9529 T; 1:1000 dilution; Cell Signaling Technology), rabbit anti-CDC25C (4688 T; 1:1000 dilution; Cell Signaling Technology), rabbit anti-cyclin A (sc-751; 1:200 dilution; Santa Cruz Biotechnology), mouse anti-cyclin B (sc-245; 1:250 dilution; Santa Cruz Biotechnology), mouse anti-cyclin D (sc-246; 1:200 dilution; Santa Cruz Biotechnology), mouse anti-cyclin E (sc-247; 1:250 dilution; Santa Cruz Biotechnology), rabbit anti-phospho-CDK1 (phospho-Thr161; 9114 S; 1:1000 dilution; Cell Signaling Technology), rabbit anti-phospho-CDK2 (phospho-Thr160; 2561 S; 1:1000 dilution; Cell Signaling Technology), rabbit anti-p21 (2947 S; 1:1000 dilution; Cell Signaling Technology), rabbit anti-p16 (ab108349; 1:1000 dilution; Abcam), rabbit anti-laminA (ab26300; 1:1000 dilution; Abcam), and mouse anti-GAPDH (G041; 1:1000 dilution; ABM). Protein expression was quantified by densitometric analysis with ImageQuant TL 8.1 software.

**Measurement of mitochondrial function**. To quantify mitochondrial shape and mass, cells were incubated in medium containing 30 nM MitoTracker Deep Red (M22426; Thermo Fisher) for 30 min at 37 °C. To quantify ROS production, cells were incubated in medium containing 5 μM MitoSOX (M36008; Thermo Fisher) for 30 min at 37 °C. After two washes with PBS, the cells were trypsinized, harvested in 500 μl of PBS, and then injected into an LSR Fortessa (Beckton Dickson) flow cytometer. The signal was acquired at an excitation wavelength of 561 nm with a 615/24 nm bandpass filter (PE-Texas Red channel). To measure MMP, cells were incubated with tetramethylrhodamine methyl ester (TMRM; I34361; Thermo Fisher) for 30 min at 37 °C. The fluorescence intensity of the samples was measured as mentioned above to quantify ROS production. To measure mitochondrial biogenesis, cells at passage 45 were transduced with the pLenti6.3/V5-DEST plasmid containing the MitoTimer reporter (#50547 l, Addgene). After 5 subsequent passages, the cells were analyzed using an LSR Fortessa flow cytometer to determine the green fluorescence (excitation wavelength of 488 nm with a 530/30 nm bandpass filter, FITC channel)/red fluorescence (excitation wavelength of 561 nm with a 583/22 nm bandpass filter, PE channel) ratio.

**Transcriptome analysis**. To extract RNA with high purity, cells were lysed with QIAzol lysis reagent (#79306 Qiagen) and precipitated with isopropanol. After washing with ethanol, RNA was diluted with RNase-free water. The RNA integrity number (RIN) was checked using a Bioanalyzer 2100 with an Agilent RNA 6000 Nano Kit according to the manufacturer's standard protocol. The average RIN value was 9.1. RNA was reverse transcribed, and the resulting cDNA was amplified and labeled with Cy3-dCTP according to the standard Agilent One-Color Microarray protocol (Agilent, V6.5, 2010). The amplified cDNA was hybridized to Agilent SurePrint G3 Human Gene Expression 8x60K v3 microarray chips containing 58,202 probes for 26,083 annotated genes. The hybridized chips were scanned with an Agilent Microarray Scanner D (Agilent, Santa Clara, CA).

**qRT-PCR analysis**. Total RNA was isolated from cells as described above. RNA was then reverse transcribed to cDNA using a Transcriptor First Strand cDNA Synthesis Kit (04 896 866 01; Roche Diagnostics). PCR was conducted in a Light Cycler 480 II detection system (Roche Diagnostics) with a KAPA SYBR FAST qPCR Master Mix (2X) Kit (07959397001; Sigma). The mRNA expression levels of the target genes were calculated using the comparative cycle threshold (CT) method and normalized to those of RPS11. The primers used for PCR are listed in Supplementary Data 5.

**Identification of differentially expressed genes (DEGs)**. We identified DEGs between two different conditions using a previously reported statistical hypothesis testing method[72]. In brief, we performed Student's t-test to obtain t-statistic values and generated an empirical null distribution for the t-statistic values by (1) subjecting the samples to 10,000 random permutations, (2) computing the t-statistic values using these random permutations, and (3) applying Gaussian density estimator to the random t-statistic values. For each gene, an adjusted P-value (P) for the observed t-statistic value was calculated by a two-tailed test using the empirical distribution. Genes with $P < 0.05$ and absolute $\log_2$-fold-change > 0.58 (1.5-fold) were selected as DEGs. This method was used to define SAGs, DAGs, and SynAGs (see the main text for details).

**Functional enrichment analysis and TF enrichment analysis**. To identify cellular processes and pathways enriched with a list of genes, we performed GOBP and GO molecular function (GOMF) enrichment analyses and Kyoto Encyclopedia of Genes and Genomes (KEGG) pathway enrichment analysis using the Database for Annotation, Visualization and Integrated Discovery (DAVID) software tool[73]. GO terms and KEGG pathways with a P-value < 0.05 and count ≥ 3 were identified as enriched with the genes. To identify the major TFs that can significantly regulate

the list of target genes, we performed TF enrichment analysis using X2Kweb[74] and identified the key TFs as those with a hypergeometric $P$-value < 0.05.

**Transcriptome analysis in other models**. We previously generated a cellular senescence compendium by combining publicly available datasets obtained for six different types of senescence: replicative senescence (RS), tumor cell senescence (TCS), oncogene-induced senescence (OIS), stem cell senescence (SCS), progeria and endothelial cell senescence (ECS)[31]. These datasets were downloaded from the GEO database. Raw data from Affymetrix array platforms were normalized using the robust multiarray average (RMA) method[75]. When no raw data were available, we used preprocessed data deposited in the GEO database. Batch corrections were performed using the ComBat function in the sva R package[76]. To summarize the expression level of each gene across different datasets, we utilized an interquartile range (IQR) method in which we selected the probe ID with the largest IQR of expression values to represent the gene. Detailed information regarding sample collection and composition can be found in a previous study[31].

**ChIP**. ChIP was performed following Abcam's crosslinking ChIP protocol with an iDeal ChIP-seq Kit for Transcription Factors (C01010055; Diagenode). In brief, proteins and chromatin in cells were crosslinked with 0.75% formaldehyde for 20 min. The crosslinked chromatin was sonicated with an ultrasonic processor (Sonics; VCX 130 PB) to shear DNA into fragments of 150–300 bp. Protein A-coated beads were coupled to 2 μg of each antibody: anti-FOXM1 (sc-376471; Santa Cruz Biotechnology), anti-E2F1 (sc-193; Santa Cruz Biotechnology), mouse IgG (sc-2025; Santa Cruz Biotechnology), and rabbit IgG (2729 S; Cell Signaling Technology). The antibody-coupled beads were subsequently incubated with the lysate at 4 °C overnight. Next, the retrieved complexes were washed and eluted in buffer containing 100 mM NaHCO$_3$ and 1% SDS, and DNA was recovered by reverse crosslinking at 65 °C overnight. DNA was then extracted by ethanol precipitation. The primer sequences used for qPCR are listed in Supplementary Data 6.

**Cell cycle analysis**. Cells were harvested by trypsinization, washed with PBS, and fixed with 70% EtOH for 8 h. Next, 12.5 μl of 1 mg/ml propidium iodide (PI) and 20 μl of 2.5 mg/ml RNase A were added to 500 μl of PBS containing $1 \times 10^5$ cells and incubated for 15 min in the dark. Finally, the cells were categorized based on the total DNA content for flow cytometric analysis of cell cycle transitions within 1 h.

**Cytokine array experiment**. A Human Cytokine Antibody Array (ab133997, Abcam) was used for quantification of 42 cytokines (ENA-78, GCSF, GM-CSF, GRO, α-GRO, I-309, α IL-1, β IL-1, IL-2, IL-3, IL-4, IL-5; IL-6, IL-7, IL-8, IL-10, IL-12p40/p70, IL-13, IL-15, INF-γ, MCP-1, MCP-2, MCP-3, MCSF, MDC, MIG, δ-MIP-1, RANTES, SCF, SDF-1, TARC, TGF-β1, TNF-α, TNF-β, EGF, IGF-I, Angiogenin, Oncostatin, Thrombopoietin, VEGF, PDGF BB, and Leptin) in culture medium according to the manufacturer's instructions. Cells were grown in DMEM supplemented with 10% FBS and were then moved to FBS-free medium 1 day before the experiment to prevent interference from cytokines in FBS. After incubating the array membranes in 1× blocking buffer for 30 min at room temperature, a sample of each culture medium was placed on the membranes and incubated overnight at 4 °C on a rocking platform shaker. Following four washes in Wash Buffer I and three washes in Wash Buffer II, the membranes were incubated first in Biotin-Conjugated Anti-Cytokines overnight at 4 °C and then in HRP-Conjugated Streptavidin for 2 h at room temperature. The washed arrays were finally treated with chemiluminescence detection reagents, and images were acquired with a LAS4000 imaging system (GE Healthcare). The level of each cytokine was quantified by densitometric analysis with ImageQuant TL 8.1 software. The relative expression levels of the cytokines are listed in Supplementary Data 1.

**Sorting of cells in different cell cycle phases and senescence states**. Cells were harvested after trypsinization and washed with PBS. To separate G1-phase cells from G2/M-phase cells, cells were incubated at 37 °C with Vybrant dye (V35004; Life Technologies) for 30 min and were then sorted based on their DNA content using a FACSAria III (Becton Dickinson). Sorted cells were incubated at 37 °C in 5% CO$_2$ overnight before treatment with DMSO or KU + Y. To separate cells with different senescence states, the forward scatter (FSC) and FITC channel intensities were obtained as the measures of cell size and lipofuscin content, respectively, and cells were then sorted based on these intensities.

**RNA interference**. siRNAs targeting RAC1B (sense: GUU GGA GAA ACG UAC GGU AAG GAU A, antisense: U AUC CUU ACC GUA CGU UUC UCC AAC) and ATP6V1G1 (sense: UUU CAG CCU GAG CUU CUU CUU UG, antisense: CA AAG AAG AAG CUC AGG CUG AAA), as well as the control siRNA (SS-1003), were synthesized by Bioneer (Daejeon, Korea). Cells were transfected with siRNAs at a final concentration of 10 nM using Lipofectamine RNAiMAX (Invitrogen).

**CDI calculation**. The CDI was calculated as follows: $CDI = AB/(A \times B)$[17]. The CDI value indicates synergistic activation (>1) or inhibition (<1).

**Statistics and reproducibility**. Statistical analyses of data other than transcriptome data were performed with GraphPad Prism 9 software. The number of samples per independent experiment ($N$) and the specific statistical hypothesis testing method [$t$-test and one- or two-way analysis of variance (ANOVA)] with the post hoc correction (Tukey's correction) for each comparison are described in the legends of the corresponding figures. $P < 0.05$ was considered statistically significant for these comparisons. Data are expressed as mean ± standard deviation (s.d.) values. Statistical analyses and their significance criteria used for the transcriptome data are described above sections ("Identification of differentially expressed genes (DEGs)" and "Functional enrichment analysis and TF enrichment analysis").

**Reporting summary**. Further information on research design is available in the Nature Research Reporting Summary linked to this article.

## Data availability

The microarray dataset generated for Fig. 2a–e was deposited in the Gene Expression Omnibus (GEO) database under accession ID GSE178115. The source data behind the graphs can be found in Supplementary Data 7. Uncropped blots are available in Supplementary Information.

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

## Acknowledgements

This study was supported by the Basic Research Program (NRF-2020R1A2C2013416, NRF-2020M3A9D8038014, and NRF-2019M3A9B6066967), and the Basic Research Laboratory program (NRF-2019R1A4A1024278) through the National Research Foundation of Korea (NRF) funded by the Ministry of Science, ICT and Technology. This research was also supported by Chonnam National University R&D Program Grant for Research Chair Professor.

## Author contributions

Y.-S.L., D.H., and S.C.P. designed the research. E.J.Y., H.-J.C., J.-A.H., and S.-H.W. performed experiments. J.H.P., C.H.P., S.Y.K., and D.H. performed the bioinformatics analysis. E.J.Y., J.H.P., D.H., and Y.-S.L. wrote the article. E.J.Y., J.H.P., J.T.P., S.C.P., D.H., and Y.-S.L. interpreted and commented on the article.

## Competing interests

The authors declare no competing interests.

## Additional information

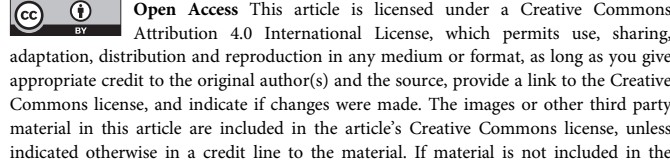

