## [Peer Review File · Communications Biology]

Reviewers' comments:

Reviewer #1 (Remarks to the Author):

The manuscript by Yang et al. presents evidence indicating that specific pharmacological inhibitors of protein kinases can reverse the growth arrest phenotype of senescent fibroblasts. Particularly the authors apply two chemical inhibitors, namely KU-60019 and Y-27632, of ATM and ROCK, respectively, to obtain this outcome. Subsequently they investigate the mechanism responsible for this results and show that the concurrent treatment with two inhibitors releases cells from FOXM1 mediated-G2 arrest and subsequently from E2F1-dependent G1 arrest through CDK-cyclin pathways activation, resulting finally in escape from senescence. This is an interesting work. Nevertheless, there are certain issues that must be more precisely addresses or defined for the message to be clear.

Comments

The term senomorphics refer to compounds that reverse many aspects of the senescent phenotype, including the SASP, while the treatment presented here focuses solely on the reversal of the growth arrest phenotype. So, it is proposed not to use the general term "senomorphic".

The two inhibitors interfere with the cell cycle control points, i.e. the G1/S and the G2/M. However, from Fig. 6 it seems that G2/M arrest is not increased in senescent cells, under basal conditions or after treatment. This must be explained. Furthermore, although it is proposed that it is the combination of the two inhibitors that leads to the release from growth arrest, however in Fig. 1b, each inhibitor alone has a significant effect. Finally, in Fig. 5C even after treatment the levels of cell cycle inhibitors p16 and p21 do not seem to change.

In many previous publications several treatments seem to release senescent cells from growth arrest, but this lasts only for a certain small period. In this work the effect of treatments is observed for only 15 days. It is important to investigate if this lasts for a much longer period. In this vein, a molecular approach, i.e. by the knock down of the targeted molecules, will indicate if this a permanent change.

Have the authors studied the expression of SASP factors after the treatment with the two inhibitors?

KU-60019 affects components of the DNA damage response pathway. Accordingly, it must be studied if the long term treatment of the cells may lead to genomic instability. In this context the authors must discuss the work by Galanos et al. [Nat Cell Boil. 18 (2016) 777-789] on the role of p53 and p21 on genomic stability and the escape from senescence.

Reviewer #2 (Remarks to the Author):

Remarks to the Author:

In this study, the authors describe the synergistic senomorphic effects of two kinase inhibitors, KU-60019 and Y-27632, in human diploid fibroblasts under replicative senescence. Activation of the cell cycle transcription factors FOXM1 and E2F1 is proposed as the mechanism behind the cooperative senomorphic effect. Senomorphic agents are relevant in the aging research field, in particular if correcting the proliferative rate and if considering the limitations of senolytic compounds. Also, combinatorial senomorphic targeting is innovative.

Although the results presented are interesting, there are over interpretations and conclusions that are not fully supported by the data. Unless the authors tone down their claims or provide additional evidence to support the conclusions, I am afraid I cannot recommend the acceptance of the manuscript for publication in its current form.

Major comments:

1. One major concern is about the misleading suggestion of 'senescence escape'. Both the abstract and the introduction sections refer to cell cycle reactivation and senescence escape. The experimental approach used does not demonstrate that senescent cells re-enter the cell cycle. The authors use late passage HDFs and they defined senescent HDFs as populations with >90% SA-b-gal cells. Thus, the senomorphic effect and restored senescence phenotypes can be merely due to the selective advantage of a small percentage of cells in the culture still able to proliferate (diluting the levels of senescent markers). The senomorphic effect in this 'fitter' subpopulation (pre-senescent) is still an interesting result.

To claim reversion, the authors would have to sort out senescent cells and demonstrate these are e.g. Ki67 negative, or alternatively test the senomorphics under serum starvation.

2. The other major concern is the sequential activation of FOXM1 and E2F1. This is unlikely to happen in individual cell cycle (in which E2F1 precedes FOXM1), but rather reflects the cell cycle profile evolution of the population. The authors should better stress this.

Fig. 6b suggests that at day 1 post-treatment there is primarily an increase in S phase cells, with very mild changes in other phases. These S phase cells expectedly must result from G1 cells (Ki67+) that get FOXM1 activation. Also, the slight decrease in G2/M can be due to FOXM1 activation previously shown to improve mitotic fitness (Laoukili et al., NCB 2005; Macedo et al., Nat Commun 2018). Fig. 6c shows that at day 8 the increase in S phase cells is even higher, again with very mild changes in other phases. This is in line with the increased proliferation rate of the cultures treated with senomorphics, with new daughter cell generations contributing to the up regulation of early cell cycle TF, E2F1.

3. Considering the concerns above, I recommend the title to be changed. Perhaps to "Co-inhibition of ATM and ROCK is a combinatorial senomorphic treatment that improves cell proliferation in replicative senescent cultures through activation of E2F1 and FOXM1 transcription factors".

4. Accordingly to the model shown in Fig. 6, the synergic effect of the drugs converges in PLK1 activation to activate FOXM1. Did the authors check the effect of PLK1 inhibitor in FOXM1 at day1 post-treatment?

5. Also, accordingly to the model, showing a pRb activation branch activating E2F1, it is intriguing why E2F1 is activated later. To strengthen the sequential activation of the TFs, it would be important to test the effect of Y+KU treatment in G1 vs. G2 sorted senescent cells (SA-g-gal+).

6. How does the absence of the specific targets ATP6V1G1 and RAC1, reported for ATMi (KU-60019) and ROCKi (Y-27632) respectively, leads to inhibition of FOXM1 and E2F1, and whether the replicative senescent cultures would recover proliferation fitness?

5. Regarding the experiment shown in Supplementary Fig. 7b, single overexpression of FOXM1 and E2F1 show equivalent effect in cell proliferation as the double overexpression. This suggests for feedback activation between both TFs but again questions about the relevance of their sequential activation.

6. Regarding the ChIP-qPCR shown in Fig. 3d, it appears that FOXM1 binds to E2F1 promoter rather weaker in comparison to other established gene targets (BIRC5, CDK1). This is not a strong argument to support the sequential activation of E2F1 by FOXM1. E2F1 might accumulate later merely due to the accumulation of fitter cycling G1 cells arising from cells dividing during the first days of treatment.

7. Fig. 6 should finish with (or only include) the model.

8. Description of statistical analysis is missing in the Materials and Methods section.

Minor comments:

1. In Fig. 1a, replace 15 day post-treatment by 15 days post-treatment.
2. In Fig. 1d, the myotracker staining shown for Y treatment is not representative of the quantitative analysis shown in the graph.
3. In Supplementary Fig. 3d the authors should alternatively measure the percentage of cells with gH2AX+53BP1 instead of number of foci per cell in n=30 cells.
4. In all legends of the main and supplementary figures, the authors should adequately indicate the statistical test applied in each analysis/graph. In Fig. 5, there is no mention to statistics. In other Figs., it is unclear if the statistic test refers to graph or all the others. Also, in all legends, the authors mention that n represents number of samples per experiment/or condition. I believe the authors refer to number of independent experiments. Please clarify this.
5. Page 7, line 139, replace Fig. 1e-f by Fig. 1d. Page 7, line 145, replace fig. 1g by Fig. 1e.
6. Page 7, line 143, the synergistic effect of Y+KU in mitochondrial features is not convincing based on CDI, at least for MMP induction.
7. Page 9, line 191, ref 24 should be replaced by refs 27-29 reporting the senomorphic effect of FOXM1.
8. Page 26, line 552, replace $\log_2\text{-fold-change} > 0.58$ (1.5-fold) by $\log_2\text{-fold-change} > 0.58$ (1.8-fold).
9. Page 3, line 51, decreases should read decrease; line 55, remove 'to reduce the deleterious effects of senescent cells', which is in redundancy.
10. Page 3, line 66, 'resulting in the G1/S transition' should read 'resulting in G1/S arrest'.

Reviewers' comments:

Reviewer #1 (Remarks to the Author):

The manuscript by Yang et al. presents evidence indicating that specific pharmacological inhibitors of protein kinases can reverse the growth arrest phenotype of senescent fibroblasts. Particularly the authors apply two chemical inhibitors, namely KU-60019 and Y-27632, of ATM and ROCK, respectively, to obtain this outcome. Subsequently they investigate the mechanism responsible for this results and show that the concurrent treatment with two inhibitors releases cells from FOXM1 mediated-G2 arrest and subsequently from E2F1-dependent G1 arrest through CDK-cyclin pathways activation, resulting finally in escape from senescence. This is an interesting work. Nevertheless, there are certain issues that must be more precisely addresses or defined for the message to be clear.

Comments

[Comment 1-1] The term senomorphics refer to compounds that reverse many aspects of the senescent phenotype, including the SASP, while the treatment presented here focuses solely on the reversal of the growth arrest phenotype. So, it is proposed not to use the general term “senomorphic”.

[Response 1-1] We agree with the reviewer on this matter, although the cotreatment of the two inhibitors has led to the changes in senescence-associated- β -galactosidase (SA- β -gal) staining, and impairment of mitochondrial or lysosomal functionality. As suggested, we thus removed the “senomorphic” term used when we refer to the cell cycle promoting function of KU-60019 and Y-27632 throughout the manuscript. For example, Abstract and Introduction were revised as follows:

Abstract:

Cellular senescence is characterized by the cessation of cell division. The multifaceted nature of cell cycle arrest necessitates the targeting of multiple factors arresting or promoting the cell cycle. We report that co-inhibition by KU-60019 and Y-27632 synergistically increases the proliferation of human diploid fibroblasts undergoing replicative senescence through activation of the transcription factors E2F1 and FOXM1 as well as mitochondrial or lysosomal impairment. Time-course transcriptome analysis identified FOXM1 and E2F1 as crucial factors promoting proliferation. Co-inhibition of the kinases ATM and ROCK first promotes the G2/M transition via FOXM1 activation, leading to accumulation of cells undergoing the G1/S transition via E2F1 activation. The combination of both inhibitors increased this effect more significantly than either inhibitor alone, suggesting synergism. Our results demonstrate a FOXM1- and E2F1-mediated molecular pathway enhancing cell cycle progression in cells with proliferative potential under replicative senescence conditions, and its anti-aging effects can be tested in vivo.

Introduction:

The progression of aging is characterized by the accumulation of senescent cells in

living organisms^{1,2}. Cells undergoing senescence exhibit a significant reduction in cell division, which is caused by suppression of the G1/S and G2/M transitions due to diverse cell cycle arrest mechanisms. For example, upregulation of p21 and p16 decreases the activity of the CDK2-cyclin E and CDK4/6-cyclin D complexes, respectively, resulting in the G1/S transition³⁻⁵. Additionally, p21 inhibits the G2/M transition by inactivating CDK1-cyclin B^{6,7}. Many anti-aging agents, such as metformin and rapamycin, have been reported to inhibit senescence-associated secretory phenotypes (SASPs)^{8,9}, but their roles in ameliorating the senescence-associated decrease in cell proliferation remain unclear. Only a few agents that can modulate cell cycle-related factors have been reported to increase the proliferation of cells undergoing senescence. For example, epigallocatechin-3-O-gallate was reported to decrease the levels of p53 and p21 and increase the number of human diploid fibroblasts (HDFs) at S-phase in replicative senescence¹⁰. In addition, curcumin was reported to increase the division of human umbilical vein endothelial cells and decrease p21 expression in oxidative stress-induced senescence¹¹.

Cellular senescence is a multifaceted process elicited by diverse senescence-associated factors, such as DNA damage, loss of proteostasis, and impairment of subcellular organelles¹². The inability of a single agent to target these different factors simultaneously is an intrinsic limitation imposed by the multifaceted nature of cellular senescence. Accordingly, the use of multiple agents, each targeting a different factor, has been suggested to collaboratively reinforce the reprogramming of the cell cycle network, thereby imparting synergism^{13,14}. However, this combinatorial targeting approach has rarely been exploited to date. Previously, we identified two compounds, an ataxia-telangiectasia mutated (ATM) inhibitor (KU-60019; KU) and a rho-associated protein kinase (ROCK) inhibitor (Y-27632; Y). KU induced the functional recovery of lysosomes by modulating the phosphorylation of a V-type ATPase subunit (ATP6V1G1)¹⁵, and Y promoted the functional recovery of the oxidative phosphorylation (OXPHOS) system in mitochondria by inhibiting the phosphorylation of mitochondrial Rac Family Small GTPase 1 (RAC1)¹⁶. However, the synergism of KU and Y, which have different modes of action, has not been investigated.

Here, we evaluated the combinatorial effect of KU and Y using HDFs as a model of replicative cellular senescence. Combination treatment led to synergism, as evidenced by the more effective suppression of senescence phenotypes, including senescence-associated cell cycle arrest, senescence-associated- β -galactosidase (SA- β -gal) staining, and impairment of mitochondrial or lysosomal functionality, even at a suboptimal concentration of Y. Time-course transcriptome analysis revealed two transcription factors (TFs)—FOXM1 and E2F1—as crucial regulators of the increase in cell proliferation. Spatiotemporal analysis of cell cycle regulators (CDKs and cyclins) further elucidated that the synergistic effect was mediated by promotion of the FOXM1-mediated G2/M transition resulting from activation of PLK1 and CDK2 via co-inhibition of ATM and ROCK, followed by accumulation of cells undergoing the E2F1-mediated G1/S transition. Therefore, our results suggest the existence of a FOXM1- and E2F1-mediated pathway enhancing cell cycle progression in cells with proliferative potential under replicative senescence conditions, and the anti-aging effects of this pathway can be tested *in vivo*.

[Comment 1-2] The two inhibitors interfere with the cell cycle control points, i.e. the G1/S and the G2/M. However, from Fig. 6 it seems that G2/M arrest is not increased in senescent cells, under basal conditions or after treatment. This must be explained. Furthermore, although

it is proposed that it is the combination of the two inhibitors that leads to the release from growth arrest, however in Fig. 1b, each inhibitor alone has a significant effect. Finally, in Fig. 5C even after treatment the levels of cell cycle inhibitors p16 and p21 do not seem to change.

[Response 1-2] Before responding to these comments, we would like to explain the modified version of our model (see Revised Fig. 6 below), thanks to the Reviewer #2's interpretation of our data. According to this modified model, co-inhibition of ATM and ROCK kinases first activates FOXM1, which promotes G2/M transition of cells which are still able to proliferate (a subpopulation at G2 phase) at early time points, as indicated by the increase of G2/M phase-specific CDK1 and cyclins A/B at 1 and 3 days post treatment (DPT; early stage), but with no significant changes in G1/S phase-specific p16 and p21 (see Revised Fig. 5a-b below). This increased G2/M transition then leads to the accumulation of cells that can undergo G1/S transition by E2F1 activation, as indicated by no changes in the levels of p16 and p21 up to 3 DPT, but the significant decrease beginning 8 DPT (late stage). Interestingly, both the abundance and phosphorylation of CDK2 and Rb were increased beginning 1 DPT (see Revised Fig. 5b and d below), suggesting that G1/S transition is ready to be activated beginning 1 DPT, but actually does not occur until later stage when CDK inhibitors (p16 and p21) become suppressed. This modified model was found consistent with the results from our new experiments suggested by Reviewer #2. In these experiments, we sorted cells into two groups of cells at G1 and G2/M, respectively, and measured proliferation of the cells in each group at 3 and 8 DPT. At 3 DPT, the cells at the G2/M group showed a much stronger increase in cell proliferation than those in the G1 group (see Revised Fig. 6d). In contrast, at 8 DPT, the cells in both the G1 and G2/M groups showed the far stronger increase in cell proliferation than those at 3 DPT, with the similar degree of the increase (see Revised Fig. 6e). These data support the temporal cell cycle progression scheme after the treatment in our model (see Revised Fig. 6f).

(Response to the 1st comment) This cell cycle progression scheme after treatment might have caused no significant increase in the count of cells at G2/M, rather a bit decrease (about 10%) at early stage (1 DPT) due to the constant progression of a subpopulation of cells to G1/S, compared to under basal conditions (DMSO-treated controls). Regarding the count of senescent cells at G2/M under basal conditions, compared to that of young cells, previous reports also showed a varying mixed pattern of an increase and no significant difference in the count of cells at G2/M, depending on the cell type and the degree of senescence (e.g., the number of passage, Reference: Zhiyong Mao et al., *Aging*, 4(6):431-5 (2012)). In our case, we think that the relatively early senescence of human diploid fibroblasts in our experimental conditions might have caused no significant difference in the count of cells as G2/M under basal conditions.

(Response to the 2nd comment) As the reviewer pointed out, the individual inhibitors (KU or Y) showed significant effects on cell proliferation, as well as other senescence phenotypes (β -galactosidase (SA- β -gal) staining, and impairment of mitochondrial or lysosomal functionality). Interestingly, however, the combination of the two inhibitors led to the much stronger effects on cell proliferation and the senescence phenotypes than each inhibitor alone, suggesting a synergistic effect (Fig. 1c and g). The main goal of our study is to elucidate a mechanistic basis for this synergistic effect. Our model describes the crosstalks of ATM and ROCK in CDK1/2 activations (CDK1-cyclin A/B via the ATM-PLK1-CDC25C

pathway and CDK2 by AKT), possibly as well as transcriptional regulation of E2F1 by FOXM1 (see Revised Fig. 6f, Pathway II-1/2/3). These crosstalks and transcriptional regulation can account for the synergistic effect of KU and Y on proliferation of senescent cells.

(Response to the 3rd comment) See the underlined above in the descriptions of our model regarding the expression patterns of p16 and p21 during the course of senescence.

To clarify these points, we added 1) the results from new experiments into Fig. 5 and 2) the modified model into Fig. 6 and also revised the corresponding sections of Results (see underlined below) to these two figures as follows.

Revised Fig. 5:

Revised Fig. 6:

Results:

KU+Y activates the CDK1-cyclin A/B pathway at the early stage and the CDK2-cyclin D/E pathway at the late stage by suppressing p16/p21-mediated inhibition

We found hyperactivation of the ATM and ROCK kinases in senescent HDFs (**Supplementary Fig. S1**), consistent with previous reports^{15,16}. ATM inhibits CDK2 activation by activating CHK2, which inhibits the CDC25A-CDK2 pathway^{15,35-37}, whereas ROCK suppresses CDK1 activation by inhibiting the AKT-PLK1-CDC25C-CDK1 pathway³⁸⁻⁴⁰. The CDK1-cyclin A/B pathway is involved in the G2/M transition, while the CDK2-cyclin A/D/E pathway, which is inhibited by p16 and p21, is involved in the G1/S transition⁴¹⁻⁴³. We next sought to determine how inhibition of ATM and ROCK by KU and Y, respectively, affects the phosphorylation or expression of these regulators. To this end, we measured changes in the phosphorylation of CHK2, TP53, AKT, PLK1, CDC25A/B/C, and CDK1/2 and the levels of CDK1/2, cyclins A/B/D/E, p16, and p21 in senescent HDFs at 1 hour and at 1, 3, and 8 days after treatment with KU, Y, and KU+Y. KU reduced the phosphorylation of CHK2 (p-Thr68), while Y increased the phosphorylation of AKT (p-Ser473) at 1 hour (**Supplementary Fig. S11**). Interestingly, KU+Y synergistically increased the levels of cyclins A and B beginning at 1 DPT and 3 DPT, respectively, consistent with the timing of FOXM1 activation, suggesting activation of the G2/M transition at the early stage after treatment (**Fig. 5a-c**). On the other hand, KU+Y did not lead to an effect on the p16 and p21 levels at the early stage (**Fig. 5b**) but decreased these levels beginning at 8 DPT (**Fig. 5d-e**); however, combination treatment synergistically increased both the abundance and phosphorylation of CDK2 and retinoblastoma (Rb) beginning at 1 DPT (**Fig. 5a-b** and **Supplementary Fig. S12**). These data suggest that the G1/S transition is ready for activation beginning at 1 DPT but is not actually activated until a later stage when the expression of the inhibitors p16 and p21 is suppressed. Moreover, our subsequent results supported this synergistic activation: 1) PLK1 activation mediated the phosphorylation of CDC25C (p-Ser198) and CHK2 inhibition mediated the dephosphorylation of CDC25A (at Ser124) and CDC25C (at Ser216) at 1 DPT (**Supplementary Fig. S13**), and 2) the phosphorylation of CDK1/2 (at Thr161 of CDK1 and at Thr160 of CDK2) increased, (**Supplementary Fig. S12**); in addition, the level of cyclin A increased beginning 1DPT, and these increase became more evident at 3 DPT (**Fig. 5a-c**).

According to previous reports, ATM also regulates CDK1 activation by inhibiting PLK1, and ROCK regulates CDK2 activation via AKT⁴⁴⁻⁵¹, suggesting that CDK1/2 may be synergistically activated by KU and Y via PLK1, a central regulator of the G2/M transition. To determine whether PLK1 can mediate the synergistic effect of KU+Y on cell proliferation, we inhibited PLK1 using volasertib and BI-2536 and measured cell proliferation at 8 DPT. As expected, inhibition of PLK1 led to significant reductions in the effects of both KU and Y (**Supplementary Fig. S14a**). PLK1 inhibition prevented the translocation of FOXM1 even more strongly after KU+Y treatment (**Supplementary Fig. S14b**), thereby inhibiting the TF activity of FOXM1. Furthermore, we also knocked down 1) the V-type ATPase subunit ATP6V1G1, a target of ATM regulated by KU¹⁵, and 2) RAC1B, a target of ROCK regulated by Y¹⁶, and monitored the alteration of KY+Y-induced nuclear translocation of FOXM1 and E2F1 by the knockdown. Interestingly, ATP6V1G1 knockdown did not affect the KU+Y-induced translocation of FOXM1 or E2F1, but RAC1B knockdown significantly inhibited the translocation of FOXM1 and E2F1 as well as the expression of E2F1 (**Supplementary Fig. S14c**). Consistently, the replicative senescent cultures of ATP6V1G1 knockdown cells recovered cell proliferation by KU+Y treatment, but that of RAC1B knockdown did not (**Supplementary Fig. S14d**). Previous our report showed that ATM-p53 axis is also involved

in senescence amelioration³⁵. In addition, RAC1B is known to promote cell proliferation and G1/S progression⁵². These data support the hypothesis that PLK1, as a potential crosstalk node of KU and Y, can mediate the synergistic effects of KU+Y on cell proliferation.

KU+Y enhances the G2/M transition at the early stage followed by the G1/S transition at the late stage

The above results suggest that KU+Y activates the FOXM1-mediated G2/M transition at the early stage after treatment and then activates the E2F1-mediated G1/S transition at the late stage. To test this hypothesis, we first performed cell cycle analysis of senescent HDFs at 1 and 8 DPT with KU, Y, or KU+Y. At the early stage (1 DPT), the proportion of G2/M-phase cells was decreased by approximately 9%, with no significant change in the proportion of G0/1-phase cells (**Fig. 6a**). In contrast, the proportion of G0/1-phase cells was reduced at the late stage (8 DPT), with no significant change in the proportion of G2/M-phase cells (**Fig. 6b**). We further sorted senescent HDFs into two groups of cells in the G1 and G2/M phases (**Fig. 6c**) and measured the proliferation of the cells in each group at 3 and 8 DPT. The cells in the G2/M group showed a much greater increase in proliferation at 3 DPT than those in the G1 group (**Fig. 6d**). In contrast, the cells in both the G1 and G2/M groups showed a far greater increase in proliferation at 8 DPT than at 3 DPT, with a similar degree of increase (**Fig. 6e**). These data collectively suggest that KU+Y first activates FOXM1, which promotes the G2/M transition in old cells with proliferative potential (a subpopulation in G2 phase) at the early stage, and the cells accumulated after the G2/M transition can then undergo the G1/S transition via E2F1 activation at the late stage when p16 and p21 expression is suppressed.

[Comment 1-3] In many previous publications several treatments seem to release senescent cells from growth arrest, but this lasts only for a certain small period. In this work the effect of treatments is observed for only 15 days. It is important to investigate if this lasts for a much longer period.

[Response 1-3] As suggested, we examined the effects of KU, Y, and KU+Y on cell proliferation at 60, 90, and 150 DPT. The results from the long-term exposure experiments showed that the synergistic effect of KU+Y on cell proliferation was maintained up to 150 DPT, comparable to that measured from the short-term exposure (15 DPT) experiments (see Revised Fig. S2 below). For other senescence phenotypes [SA- β -gal staining, lysosomal functionality (lipofuscin), cell size, and DNA damage (γ H2AX and 53BP1 foci and neutral comet assays)], the significant effects of KU, Y, and KU+Y were also observed even after the long-term exposure. However, the difference in the effects on these senescence phenotypes between KU and KU+Y has decreased a bit with a varying degree for each phenotype, compared to the difference observed after the short-term exposure (see Revised Fig. S3 below), suggesting that the synergistic effect of KU+Y on these senescence phenotypes has been attenuated, probably due to the saturated effect of KU.

To clarify these points, we added the results from new long-term exposure experiments to

Supplementary Fig. S2 and S3. We also revised the corresponding section of Results to describe the new results as follows.

Revised Supplementary Fig. S2:

Revised Supplementary Fig. S3:

Results:

“KU and Y synergistically restore the proliferation of senescent HDFs

... Finally, we examined the effects of KU, Y, and KU+Y on cell proliferation at 60, 90, and 150 DPT. After long-term exposure, the synergistic effect of KU+Y on cell proliferation was maintained up to 150 DPT and was comparable to that measured after short-term exposure (15 DPT) (**Supplementary Fig. S2c-d**). Taken together, these data suggest that inhibition of ATM and ROCK by KU+Y synergistically contributes to increasing the proliferation of senescent HDFs.

“KU and Y cooperatively suppress the senescence phenotypes of HDFs

... Moreover, we examined the long-term effects of KU, Y, and KU+Y on the above senescence phenotypes (SA- β -gal staining, lysosomal function, and cell size) at 60 and 90 DPT. Significant effects of KU, Y, and KU+Y on these phenotypes were observed even after long-term exposure. However, the differences in the effects of KU and KU+Y decreased slightly to varying degrees for each phenotype compared to those observed after short-term exposure (**Supplementary Fig. S3d-g**), suggesting that the synergistic effects of KU+Y on these senescence phenotypes were attenuated, probably due to the saturation effect of KU. ...

...

... To examine the long-term effect on DNA damage, we further performed neutral comet assays and counted γ H2AX and 53BP1 foci at 60 DPT in senescent HDFs treated with KU, Y, or KU+Y. Even after long-term exposure (60 DPT), reduced DNA damage was still observed in the treated cells compared to the DMSO-treated control cells (**Supplementary Fig. S3h-i**). On the other hand, compared to short-term exposure, long-term exposure increased the absolute levels of DNA damage. This increased level of DNA damage in the long-term treated cells was consistent with the lack of a significant decrease in p16 and p21 expression (**Supplementary Fig. S3j**). Of note, this expression pattern of p21 would be in line with the previously reported role of p21 in the induction of genomic instability¹⁸. ...

[Comment 1-4] In this vein, a molecular approach, i.e. by the knock down of the targeted molecules, will indicate if this a permanent change.

[Response 1-4] As suggested (also by Reviewer #2), we inhibited PLK1, which can mediate the crosstalk of KU and Y according to our model (see Pathway II-1 in Revised Fig. 6f in **Response 1-2** above) and the previous literature (ATM also regulates CDK1 activation by inhibiting PLK1, and ROCK regulates CDK2 activation via AKT⁴⁴⁻⁵¹). As expected, the inhibition of PLK1 led to significant reductions in the effects of both KU and Y (see Revised Supplementary Fig. S14a-b below), suggesting the presence of a PLK1-mediated crosstalk between KU and Y.

Furthermore, we knocked down a vATPase subunit ATP6V1G1 (a target of ATM regulated by KU) and RAC1B (a target of ROCK regulated by Y). After the knockdown, we then monitored the alteration of KY+Y-induced nuclear translocation of FOXM1 and E2F1 by the knockdown

at 8 DPT. Interestingly, ATP6V1G1 knockdown did not affect the KU+Y-induced translocation of FOXM1 or E2F1, but RAC1B knockdown significantly inhibited the translocation of FOXM1 and E2F1 as well as the expression of E2F1 (Supplementary Fig. S14c). Consistently, the replicative senescent cultures of ATP6V1G1 knockdown cells recovered cell proliferation by KU+Y treatment, but that of RAC1B knockdown did not (Supplementary Fig. S14d). Previous our report showed that ATM-p53 axis is also involved in senescence amelioration³⁵. In addition, RAC1B is known to promote cell proliferation and G1/S progression⁵².

To clarify these points, we added the results from new knockdown and inhibition experiments to Supplementary Fig. S14 and revised the corresponding section of Results to describe the results as follows.

Revised Supplementary Fig. S14

Results:

KU+Y activates the CDK1-cyclin A/B pathway at the early stage and the CDK2-cyclin D/E pathway at the late stage by suppressing p16/p21-mediated inhibition

... According to previous reports, ATM also regulates CDK1 activation by inhibiting PLK1, and ROCK regulates CDK2 activation via AKT⁴⁴⁻⁵¹, suggesting that CDK1/2 may be synergistically activated by KU and Y via PLK1, a central regulator of the G2/M transition. To determine whether PLK1 can mediate the synergistic effect of KU+Y on cell proliferation, we inhibited PLK1 using volasertib and BI-2536 and measured cell proliferation at 8 DPT. As expected, inhibition of PLK1 led to significant reductions in the effects of both KU and Y (**Supplementary Fig. S14a**). PLK1 inhibition prevented the translocation of FOXM1 even more strongly after KU+Y treatment (**Supplementary Fig. S14b**), thereby inhibiting the TF activity of FOXM1. Furthermore, we also knocked down 1) the V-type ATPase subunit ATP6V1G1, a target of ATM regulated by KU¹⁵, and 2) RAC1B, a target of ROCK regulated by Y¹⁶, and monitored the alteration of KY+Y-induced nuclear translocation of FOXM1 and E2F1 by the knockdown at 8 DPT. Interestingly, ATP6V1G1 knockdown did not affect the KU+Y-induced translocation of FOXM1 or E2F1, but RAC1B knockdown significantly inhibited the translocation of FOXM1 and E2F1 as well as the expression of E2F1 (**Supplementary Fig. S14c**). Consistently, the replicative senescent cultures of ATP6V1G1 knockdowned cells recovered cell proliferation by KU+Y treatment, but that of RAC1B knockdowned did not (**Supplementary Fig. S14d**). Previous our report showed that ATM-p53 axis is also involved in senescence amelioration³⁵. In addition, RAC1B is known to promote cell proliferation and G1/S progression⁵². These data support the hypothesis that PLK1, as a potential crosstalk node of KU and Y, can mediate the synergistic effects of KU+Y on cell proliferation.

[Comment 1-5] Have the authors studied the expression of SASP factors after the treatment with the two inhibitors?

[Response 1-5] As suggested, we measured the levels of cytokines in senescent cells treated with KU, Y, or KU+Y at 8 DPT using Abcam Human cytokine antibody array containing antibodies against 42 cytokines and then compared these levels with those in DMSO-treated controls. Among the 42 cytokines, 38 tended to be upregulated after the treatment while 4 tended to be downregulated (see Revised Fig. 1f below). KU, Y, or KU+Y appeared to restore these changes in senescent HDFs toward the levels in young HDFs. However, the differences in log₂-fold-changes between 1) KU or Y and 2) KU+Y seem not significant, suggesting that the synergistic effects of KU+Y on SASPs might be not apparent.

To clarify this point, we added the results from the cytokine array to Fig. 1f and revised the corresponding section of Results as follows.

Revised Fig. 1f:

Results:

“KU and Y cooperatively suppress the senescence phenotypes of HDFs

... To examine the synergistic effect of KU+Y on SASPs, we next measured the levels of cytokines in senescent cells treated with KU, Y, or KU+Y at 8 DPT using an Abcam Human Cytokine Antibody Array, which contains antibodies against 42 cytokines, and compared these levels with those in DMSO-treated control cells. Of the 42 cytokines, 38 showed a trend toward upregulation and 4 showed a trend toward downregulation after treatment (Fig. 1f). KU, Y, and KU+Y appeared to restore these levels in senescent HDFs toward the levels in young HDFs. However, the differences in the log₂-fold-change values between KU and KU+Y and between Y and KU+Y were not significant, suggesting that the synergistic effects of KU+Y on SASPs might not be apparent. ...

[Comment 1-6] KU-60019 affects components of the DNA damage response pathway. Accordingly, it must be studied if the long term treatment of the cells may lead to genomic instability. In this context the authors must discuss the work by Galanos et al. [Nat Cell Boil. 18 (2016) 777-789] on the role of p53 and p21 on genomic stability and the escape from senescence.

[Response 1-6] As suggested, we performed neutral comet assays and counted γ H2AX and 53BP1 foci to quantitatively measure DNA damages at 15 and 60 DPT in senescent HDFs treated with KU, Y, or KU+Y (see Revised Fig. S3h-j below). The long-term exposure (60 DPT)

still showed the reduced DNA damage in the treated cells, compared to that in DMSO-treated control cells (**Supplementary Fig. S3h-i**). On the other hand, the long-term exposure increased the absolute levels of DNA damage compared to the short-term exposure. This increased level of DNA damage in the long-term treated cells was consistent with no significant decrease of p16 and p21 (**Supplementary Fig. S3j**). Of note, this expression pattern of p21 is in line with the role of p21 in induction of genomic instability previously reported by Galanos et al.

To clarify these points, we added the results from new comet and γ H2AX and 53BP1 foci assays to Supplementary Fig. S3h-j and revised the corresponding section of Results as follows.

Revised Supplementary Fig. S3h-j:

Results:

“KU and Y cooperatively suppress the senescence phenotypes of HDFs

... We also examined the effect of cotreatment on DNA damage and found that cotreatment led to greater reductions in the numbers of γ H2AX and 53BP1 foci as well as in the length of comet tails originating from DNA double-strand breaks at 15 DPT than did either single-drug treatment (**Supplementary Fig. S3h-i**). To examine the long-term effect on DNA damage, we further performed neutral comet assays and counted γ H2AX and 53BP1 foci at 60 DPT in senescent HDFs treated with KU, Y, or KU+Y. Even after long-term KU exposure (60 DPT), reduced DNA damage was still observed in the treated cells compared to the DMSO-treated control cells (**Supplementary Fig. S3h-i**). On the other hand, compared to short-term exposure, long-term exposure increased the absolute levels of DNA damage. This increased level of DNA damage in the long-term treated cells was consistent with the lack of a significant decrease in p16 and p21 expression (**Supplementary Fig. S3j**). Of note, this expression pattern of p21 would be in line with the previously reported role of p21 in the induction of genomic instability¹⁸. ...

Reviewer #2 (Remarks to the Author):

Remarks to the Author:

In this study, the authors describe the synergistic senomorphic effects of two kinase inhibitors, KU-60019 and Y-27632, in human diploid fibroblasts under replicative senescence. Activation of the cell cycle transcription factors FOXM1 and E2F1 is proposed as the mechanism behind the cooperative senomorphic effect. Senomorphic agents are relevant in the aging research field, in particular if correcting the proliferative rate and if considering the limitations of senolytic compounds. Also, combinatorial senomorphic targeting is innovative.

Although the results presented are interesting, there are over interpretations and conclusions that are not fully supported by the data. Unless the authors tone down their claims or provide additional evidence to support the conclusions, I am afraid I cannot recommend the acceptance of the manuscript for publication in its current form.

Major comments:

[Comment 2-1] One major concern is about the misleading suggestion of ‘senescence escape’. Both the abstract and the introduction sections refer to cell cycle reactivation and senescence escape. The experimental approach used does not demonstrate that senescent cells re-enter the cell cycle. The authors use late passage HDFs and they defined senescent HDFs as populations with >90% SA-b-gal cells. Thus, the senomorphic effect and restored senescence phenotypes can be merely due to the selective advantage of a small percentage of cells in the culture still able to proliferate (diluting the levels of senescent markers). The senomorphic effect in this ‘fitter’ subpopulation (pre-senescent) is still an interesting result.

To claim reversion, the authors would have to sort out senescent cells and demonstrate these are e.g. Ki67 negative, or alternatively test the senomorphics under serum starvation.

[Response 2-1] To address the 1st issue regarding cell cycle reactivation (or reversion) and senescence escape, we revised the entire manuscript to remove all these terminologies. For example, see our revised Abstract and Introduction in **Response 1-1** above.

To address the 2nd issue regarding the ‘fitter’ subpopulation, we would like to thank Reviewer 2 for helping us interpret our results appropriately. Following Reviewer 2’s comment, we revised our model as follow:

Co-inhibition of ATM and ROCK kinases first activates FOXM1, which promotes G2/M transition of cells which are still able to proliferate (a subpopulation at G2 phase) at early time points, as indicated by the increase of G2/M phase-specific CDK1 and cyclins A/B at 1 and 3 days post treatment (DPT; early stage), but with no significant changes in G1/S phase-specific p16 and p21 (see Revised Fig. 5a-b below). This increased G2/M transition then leads to the accumulation of cells that can undergo G1/S transition by E2F1 activation, as indicated by no changes in the levels of p16 and p21 up to 3 DPT, but the significant decrease beginning 8 DPT (late stage). Interestingly, both the abundance and phosphorylation of CDK2 and Rb were increased beginning 1 DPT (see Revised Fig. 5a and Supplementary Fig. S12 below), suggesting that G1/S transition is ready to be activated beginning 1 DPT, but actually does not occur until later stage when CDK inhibitors (p16 and p21) become suppressed. This modified

model was found consistent with the results from our new experiments suggested by Reviewer #2 (see **Comment 2-5** below). In these experiments, we sorted cells into two groups of cells at G1 and G2/M, respectively, and measured proliferation of the cells in each group at 3 and 8 DPT. At 3 DPT, the cells at the G2/M group showed a much stronger increase in cell proliferation than those in the G1 group (see Revised Fig. 6d). In contrast, at 8 DPT, the cells in both the G1 and G2/M groups showed the far stronger increase in cell proliferation than those at 3 DPT, with the similar degree of the increase (see Revised Fig. 6e). These data support the temporal cell cycle progression scheme after the treatment in our model (see Revised Fig. 6f).

To address the 3rd issue regarding the additional experiments requested to claim the reversion, we found that Ki67 was predominantly localized to the nucleus in young HDFs while it was to the cytoplasm in senescent HDFs (see Figure A below). Concerned with this difference in the intracellular distribution of Ki67, we sorted senescent HDFs, rather than Ki67 expression, based on 1) cell size and 2) auto-fluorescence from lipofuscin, both of which are supposed to be increased in senescent cells. We then isolated two groups of senescent cells with low (P3) and high levels (P1) of these two measures (see Figure B, left) and compared the effects of KU+Y on cell proliferation at 8 DPT between P1 and P3 groups. The cells in the P1 group (with relatively higher senescence characteristics) showed a weaker effect of KU+Y on the improvement of cell proliferation than those in the P3 group (see Figure B, right). This reduced effect in the P1 group, expected to have stronger cell cycle arrest, supports that Reviewer 2's interpretation is correct. Moreover, we examined the effect of KU, Y, and KU+Y on proliferation of cells at 8 DPT after incubating the cells for 8 days under the serum starvation condition. No improvement of cell proliferation was observed after drug treatment (see Figure D below), which supports again Reviewer 2's interpretation. The same results were observed when the cells were treated with hydroxyurea (G1 arrest inducer) and nocodazole (G2 arrest inducer). Accordingly, we removed all the misleading terminologies from the manuscript and revised our model in accordance with Reviewer 2's interpretation, as described above.

Revised Fig. 5a-b:

Revised Supplementary Fig. S12:

a

b

c

Additional Figures for [Response 2-1]:

[Comment 2-2] 2. The other major concern is the sequential activation of FOXM1 and E2F1. This is unlikely to happen in individual cell cycle (in which E2F1 precedes FOXM1), but rather reflects **the cell cycle profile evolution of the population**. The authors should better stress(address) this.

Fig. 6b suggests that at day 1 post-treatment there is primarily an increase in S phase cells, with

very mild changes in other phases. These S phase cells expectedly must result from G1 cells (Ki67+) that get FOXM1 activation. Also, the slight decrease in G2/M can be due to FOXM1 activation previously shown to improve mitotic fitness (Laoukili et al., NCB 2005; Macedo et al., Nat Communics 2018). Fig. 6c shows that at day 8 the increase in S phase cells is even higher, again with very mild changes in other phases. This is in line with the increased proliferation rate of the cultures treated with senomorphics, with new daughter cell generations contributing to the up regulation of early cell cycle TF, E2F1.

[Response 2-2] As described in our **Response 2-1**, we agree with Reviewer #2's interpretation and revised our model and interpretation of cell cycle experiment results accordingly. After comparing the effects of KU+Y on cell proliferation in two groups of cells at G1 and G2/M phases, we revised Fig. 6 and the corresponding section of Results as follows.

Revised Fig. 6:

Results:

KU+Y enhances the G2/M transition at the early stage followed by the G1/S transition at the late stage

The above results suggest that KU+Y activates the FOXM1-mediated G2/M transition at the early stage after treatment and then activates the E2F1-mediated G1/S transition at the late stage. To test this hypothesis, we first performed cell cycle analysis of senescent HDFs at 1 and 8 DPT with KU, Y, or KU+Y. At the early stage (1 DPT), the proportion of G2/M-phase cells was decreased by approximately 9%, with no significant change in the proportion of G0/1-phase cells (**Fig. 6a**). In contrast, the proportion of G0/1-phase cells was reduced at the late stage (8 DPT), with no significant change in the proportion of G2/M-phase cells (**Fig. 6b**). We further sorted senescent HDFs into two groups of cells in the G1 and G2/M phases (**Fig. 6c**) and measured the proliferation of the cells in each group at 3 and 8 DPT. The cells in the G2/M group showed a much greater increase in proliferation at 3 DPT than those in the G1 group (**Fig. 6d**). In contrast, the cells in both the G1 and G2/M groups showed a far greater increase in proliferation at 8 DPT than at 3 DPT, with a similar degree of increase (**Fig. 6e**). These data collectively suggest that KU+Y first activates FOXM1, which promotes the G2/M transition in old cells with proliferative potential (a subpopulation in G2 phase) at the early stage, and the cells accumulated after the G2/M transition can then undergo the G1/S transition via E2F1 activation at the late stage when p16 and p21 expression is suppressed.

[Comment 2-3] Considering the concerns above, I recommend the title to be changed. Perhaps to “Co-inhibition of ATM and ROCK is a combinatorial senomorphic treatment that improves cell proliferation in replicative senescent cultures through activation of E2F1 and FOXM1 transcription factors”.

[Response 2-3] As suggested, we changed the title into “Co-inhibition of ATM and ROCK synergistically improves cell proliferation in replicative senescence by activating FOXM1 and E2F1”. Due to the word limit according to the journal policy, we modified a bit the suggested title to reduce the word count.

[Comment 2-4] Accordingly to the model shown in Fig. 6, the synergic effect of the drugs converges in PLK1 activation to activate FOXM1. Did the authors check the effect of PLK1 inhibitor in FOXM1 at day1 post-treatment?

[Comment 2-6] How does the absence of the specific targets ATP6V1G1 and RAC1, reported for ATMi (KU-60019) and ROCKi (Y-27632) respectively, leads to inhibition of FOXM1 and E2F1, and whether the replicative senescent cultures would recover proliferation fitness?

[Response 2-4 & 6] As suggested, we inhibited PLK1, which can mediate the crosstalk of KU and Y according to our model (see Pathway II-1 in Revised Fig. 6f in **Response 2-2** above) and the previous literature (ATM also regulates CDK1 activation by inhibiting PLK1, and ROCK regulates CDK2 activation via AKT⁴⁴⁻⁵¹). As expected, the inhibition of PLK1 led to significant

reductions in the effects of both KU and Y (see Revised Supplementary Fig. S14a-b below), suggesting the presence of a PLK1-mediated crosstalk between KU and Y.

Furthermore, we knocked down a vATPase subunit ATP6V1G1 (a target of ATM regulated by KU) and RAC1B (a target of ROCK regulated by Y). After the knockdown, we then monitored the alteration of KY+Y-induced nuclear translocation of FOXM1 and E2F1 by the knockdown at 8 DPT. Interestingly, ATP6V1G1 knockdown did not affect the KU+Y-induced translocation of FOXM1 or E2F1, but RAC1B knockdown significantly inhibited the translocation of FOXM1 and E2F1 as well as the expression of E2F1 (Supplementary Fig. S14c). Consistently, the replicative senescent cultures of ATP6V1G1 knockdowned cells recovered cell proliferation by KU+Y treatment, but that of RAC1B knockdowned did not (Supplementary Fig. S14d). Previous our report showed that ATM-p53 axis is also involved in senescence amelioration³⁵. In addition, RAC1B is known to promote cell proliferation and G1/S progression⁵².

To clarify these points, we added the results from new knockdown and inhibition experiments to Supplementary Fig. S14 and revised the corresponding section of Results to describe the results as follows.

Revised Supplementary Fig. S14

Results:

KU+Y activates the CDK1-cyclin A/B pathway at the early stage and the CDK2-cyclin D/E pathway at the late stage by suppressing p16/p21-mediated inhibition

... According to previous reports, ATM also regulates CDK1 activation by inhibiting PLK1, and ROCK regulates CDK2 activation via AKT⁴⁴⁻⁵¹, suggesting that CDK1/2 may be synergistically activated by KU and Y via PLK1, a central regulator of the G2/M transition. To determine whether PLK1 can mediate the synergistic effect of KU+Y on cell proliferation, we inhibited PLK1 using volasertib and BI-2536 and measured cell proliferation at 8 DPT. As expected, inhibition of PLK1 led to significant reductions in the effects of both KU and Y (**Supplementary Fig. S14a**). PLK1 inhibition prevented the translocation of FOXM1 even more strongly after KU+Y treatment (**Supplementary Fig. S14b**), thereby inhibiting the TF activity of FOXM1. Furthermore, we also knocked down 1) the V-type ATPase subunit ATP6V1G1, a target of ATM regulated by KU¹⁵, and 2) RAC1B, a target of ROCK regulated by Y¹⁶, and monitored the alteration of KY+Y-induced nuclear translocation of FOXM1 and E2F1 by the knockdown at 8 DPT. Interestingly, ATP6V1G1 knockdown did not affect the KU+Y-induced translocation of FOXM1 or E2F1, but RAC1B knockdown significantly inhibited the translocation of FOXM1 and E2F1 as well as the expression of E2F1 (**Supplementary Fig. S14c**). Consistently, the replicative senescent cultures of ATP6V1G1 knockdowned cells recovered cell proliferation by KU+Y treatment, but that of RAC1B knockdowned did not (**Supplementary Fig. S14d**). Previous our report showed that ATM-p53 axis is also involved in senescence amelioration³⁵. In addition, RAC1B is known to promote cell proliferation and G1/S progression⁵². These data support the hypothesis that PLK1, as a potential crosstalk node of KU and Y, can mediate the synergistic effects of KU+Y on cell proliferation.

[Comment 2-5] Also, accordingly to the model, showing a pRb activation branch activating E2F1, it is intriguing why E2F1 is activated later. To strengthen the sequential activation of the TFs, it would be important to test the effect of Y+KU treatment in G1 vs. G2 sorted senescent cells (SA-g-gal+).

[Response 2-5] See the underlined in our Response 2-1 above. Briefly, we sorted cells into two groups of cells at G1 and G2/M, respectively, and measured proliferation of the cells in each group at 3 and 8 DPT. At 3 DPT, the cells at the G2/M group showed a much stronger increase in cell proliferation than those in the G1 group (see Revised Fig. 6d). In contrast, at 8 DPT, the cells in both the G1 and G2/M groups showed the far stronger increase in cell proliferation than those at 3 DPT, with the similar degree of the increase (see Revised Fig. 6e in Response 2-2 above). These data support the temporal cell cycle progression scheme after the treatment in our model (see Revised Fig. 6f in Response 2-2).

[Comment 2-7] Regarding the experiment shown in Supplementary Fig. 7b, single overexpression of FOXM1 and E2F1 show equivalent effect in cell proliferation as the double

overexpression. This suggests for feedback activation between both TFs but again questions about the relevance of their sequential activation.

[Response 2-7] We agree with reviewer 2 on this matter. In our system, a subpopulation of cells at G2 phase were first affected by KU+Y via activation of FOXM1, leading to the accumulation of cells that can undergo G1/S transition by E2F1 activation. However, the opposite can be also true. Upon overexpression of E2F1, the triggering cell population can be the cells undergoing G1/S transition by E2F1 activation, leading to the accumulation of cells that can undergo G2/M transition. Thus, these data indicate a feedback activation between the two TFs, as the reviewer pointed out. We incorporated this aspect to our modified model.

[Comment 2-8] Regarding the ChIP-qPCR shown in Fig. 3d, it appears that FOXM1 binds to E2F1 promoter rather weaker in comparison to other established gene targets (BIRC5, CDK1). This is not a strong argument to support the sequential activation of E2F1 by FOXM1. E2F1 might accumulate later merely due to the accumulation of fitter cycling G1 cells arising from cells dividing during the first days of treatment.

[Response 2-8] As described in our responses above, we agree with reviewer 2's interpretation and revised our model and interpretation of cell cycle experiment results accordingly (see **Response 2-1 and 2-2** above).

[Comment 2-9] Fig. 6 should finish with (or only include) the model.

[Response 2-9] As suggested, Fig. 6 was revised to finish with the modified model (see Fig. 6f in **Response 2-2** above).

[Comment 2-10] Description of statistical analysis is missing in the Materials and Methods section.

[Response 2-10] As suggested, detailed descriptions for statistical analysis were added to Materials and Methods.

Materials and Methods:

Statistical analyses

Statistical analyses and their significance criteria used for the transcriptome data are described above. Statistical analyses of data other than transcriptome data were performed with GraphPad Prism 9 software. The number of samples per independent experiment (N) and the specific statistical hypothesis testing method [t test and one- or two-way analysis of variance

(ANOVA)] with the post hoc correction (Tukey's correction) for each comparison are described in the legends of the corresponding figures. $P < 0.05$ was considered statistically significant for these comparisons.

Minor comments:

1. In Fig. 1a, replace 15 day post-treatment by 15 days post-treatment.

[Response 2-11] Corrected as suggested (see below)

(Previous Fig. 1a)

(Revised Fig. 1a)

a

a

2. In Fig. 1d, the myotracker staining shown for Y treatment is not representative of the quantitative analysis shown in the graph.

[Response 2-12] Replaced the image as the one representing the quantification results as shown below (see 'Y' image).

(Previous Fig. 1d)

d

(Revised Fig. 1d)

f

3. In Supplementary Fig. 3d the authors should alternatively measure the percentage of cells with γ H2AX+53BP1 instead of number of foci per cell in n=30 cells.

[Response 2-11] Corrected as suggested (see below)

(Previous Fig. S3d)

(Revised Fig. S4h)

4. In all legends of the main and supplementary figures, the authors should adequately indicate the statistical test applied in each analysis/graph. In Fig. 5, there is no mention to statistics. In other Figs., it is unclear if the statistic test refers to graph or all the others. Also, in all legends, the authors mention that n represents number of samples per experiment/or condition. I believe the authors refer to number of independent experiments. Please clarify this.

[Response 2-14] As suggestion, we clarified the number of independent experiments (N) and statistical test used for the comparison in the legend of the corresponding figure. For example, see underlined below:

Figure 1. Synergistic effects of KU+Y on senescence phenotypes. (a) Screening for the optimal concentrations of KU and Y based on their individual effects on cell proliferation (CCK absorbance). Senescent HDFs were treated with KU (0.125, 0.25, 0.5, and 1.5 μ M) and Y (2.5, 5, 7.5, and 10 μ M) for 15 days. Eight independent experiments were conducted per condition (N=8). (b) Effects of Y, KU and KU+Y on cell proliferation (N=11), SA- β -gal staining (N=11), and lipofuscin staining (N=3) at 8 DPT. The data are shown as the mean \pm standard deviation (s.d.) values. **, $P < 0.01$; ***, $P < 1.0 \times 10^{-3}$; ****, $P < 1.0 \times 10^{-4}$ by one-way ANOVA with Tukey's post hoc correction.

Figure 3. FOXM1 as a regulator of cell proliferation activated at the early stage after treatment. (a) Attenuated proliferation of KU+Y-treated senescent HDFs by FOXM1 knockdown (top) and immunoblots of FOXM1 in senescent HDFs after FOXM1 knockdown (bottom). Cell proliferation was measured at 8 DPT. The data are shown as the mean \pm s.d. values; N = 10 per condition. ... For statistical analyses, (a-c): *, $P < 0.05$; ***, $P < 1.0 \times 10^{-3}$; ****, $P < 1.0 \times 10^{-4}$ by one-way ANOVA with Tukey's post hoc correction. (d): *, $P < 0.05$; ***, $P < 1.0 \times 10^{-3}$; ****, $P < 1.0 \times 10^{-4}$ by two-way ANOVA with Tukey's post hoc correction.

Figure 4. E2F1 as a key regulator of cell proliferation at the late stage after treatment. ... For statistical analyses, (a-c): **, $P < 0.01$; ***, $P < 1.0 \times 10^{-3}$; ****, $P < 1.0 \times 10^{-4}$ by one-way ANOVA with Tukey's post hoc correction. (d): *, $P < 0.05$; **, $P < 0.01$; ***, $P < 1.0 \times 10^{-3}$; ****, $P < 1.0 \times 10^{-4}$ by two-way ANOVA with Tukey's post hoc correction.

Figure 5. Induction of CDK-cyclin pathway activation by KU+Y at the early and late stages after treatment. (a-b and d) Representative immunoblots of the indicated cell cycle regulators at 1 DPT (a), 3 DPT (b) or 8 DPT (d) with KU, Y, or KU+Y. The arrows indicate the bands representing the phosphorylated forms of the proteins. (c and e) The protein levels of cyclins (c), P21, and P16 (e, bottom) and the levels of phosphorylated RB and P53 (e, top) were quantified using images acquired from N = 2-4 samples per experiment. The data are shown as the mean \pm s.d. values. *, $P < 0.05$; **, $P < 0.01$; ***, $P < 1.0 \times 10^{-3}$; ****, $P < 1.0 \times 10^{-4}$ by one-way ANOVA with Tukey's post hoc correction.

5. Page 7, line 139, replace Fig. 1e-f by Fig. 1d. Page 7, line 145, replace fig. 1g by Fig. 1e.

[Response 2-15] Corrected as follows:

(Previous manuscript)

control. As expected, cotreatment led to stronger induction of mitochondrial fragmentation in senescent HDFs than did either single-drug treatment, restoring mitochondrial fragmentation to a level similar to that observed in young HDFs (**Fig. 1e-f**). Additionally, several key features related to mitochondrial functions were improved by cotreatment, as indicated by the decreases in total mitochondrial mass and ROS production, as well as the increases in mitochondrial biogenesis and mitochondrial membrane potential (MMP) (**Supplementary Fig. S4a-d**). The temporal CDI profiles confirmed the synergistic effects of KU and Y on these mitochondrial function-related features from 3 to 15 DPT (**Fig. 1g**). Taken together, these data suggest that

(Revised manuscript)

functionality of mitochondrial proteins may improve mitochondrial quality control. As expected, cotreatment led to stronger induction of mitochondrial fragmentation in senescent HDFs than did either single-drug treatment, restoring mitochondrial fragmentation to a level similar to that observed in young HDFs (**Fig. 1d**). Additionally, several key features related to mitochondrial function were improved by cotreatment, as indicated by the decreases in total mitochondrial mass and ROS production, as well as the increases in mitochondrial biogenesis and mitochondrial membrane potential (MMP) (**Supplementary Fig. S4a-d**). The temporal CDI profiles confirmed the synergistic effects of KU and Y on the reductions in mitochondrial mass and ROS production and the induction of mitochondrial biogenesis from 3 to 15 DPT, but no synergism on MMP (**Fig. 1e**). **To examine the synergistic effect of KU+Y on SASPs, we**

6. Page 7, line 143, the synergistic effect of Y+KU in mitochondrial features is not convincing based on CDI, at least for MMP induction.

[Response 2-16] Corrected as follows:

Additionally, several key features related to mitochondrial function were improved by cotreatment, as indicated by the decreases in total mitochondrial mass and ROS production, as well as the increases in mitochondrial biogenesis and mitochondrial membrane potential (MMP) (**Supplementary Fig. S4a-d**). The temporal CDI profiles confirmed the synergistic effects of KU and Y on the reductions in mitochondrial mass and ROS production and the induction of mitochondrial biogenesis from 3 to 15 DPT, but no synergism on MMP (**Fig. 1e**).

7. Page 9, line 191, ref 24 should be replaced by refs 27-29 reporting the senomorphic effect of FOXM1.

[Response 2-17] Corrected as follows:

Notably, a cell cycling promoting function of FOXM1 was previously reported²³⁻²⁵

References

- 23 Macedo, J. C. *et al.* FoxM1 repression during human aging leads to mitotic decline and aneuploidy-driven full senescence. *Nat Commun* **9**, 2834, doi:10.1038/s41467-018-05258-6 (2018).
- 24 Smirnov, A. *et al.* FOXM1 regulates proliferation, senescence and oxidative stress in keratinocytes and cancer cells. *Aging (Albany NY)* **8**, 1384-1397, doi:10.18632/aging.100988 (2016).
- 25 Golson, M. L. *et al.* Activation of FoxM1 Revitalizes the Replicative Potential of Aged beta-Cells in Male Mice and Enhances Insulin Secretion. *Diabetes* **64**, 3829-3838, doi:10.2337/db15-0465 (2015).

8. Page 26, line 552, replace log2-fold-change>0.58 (1.5-fold) by log2-fold-change>0.58(1.8-fold).

[Response 2-18] We think that the reviewer misunderstood 0.58 with 0.85. $\text{Log}_2(1.5) = 0.58$ [or $2^{0.58} = 1.5$] and $2^{(0.85)} = 1.80$. Thus, we kept the original description.

9. Page 3, line 51, decreases should read decrease; line 55, remove ‘to reduce the deleterious effects of senescent cells’, which is in redundancy.

[Response 2-19] For “decrease” in line 51, we revised the sentence for clarification as follows:

For example, upregulation of p21 and p16 decreases the activity of the CDK2-cyclin E and CDK4/6-cyclin D complexes, respectively, resulting in the G1/S transition³⁻⁵

For the ‘to reduce the deleterious effects of senescent cells’, this description was removed as we revising the introduction to remove the terminologies like cell cycle reactivation and senomorphism-related terms (see the revised Introduction in Response 1-1 above).

10. Page 3, line 66, ‘resulting in the G1/S transition’ should read ‘resulting in G1/S arrest’.

[Response 2-20] Corrected as suggested:

For example, upregulation of p21 and p16 decreases the activity of the CDK2-cyclin E and CDK4/6-cyclin D complexes, respectively, resulting in the G1/S transition³⁻⁵

REVIEWERS' COMMENTS:

Reviewer #1 (Remarks to the Author):

The authors have properly addressed all issues raised and therefore, I recommend publication.

Reviewer #2 (Remarks to the Author):

The authors have addressed all the major concerns raised by the Reviewers and the Editor. Substantial amount of new data have improved the quality of the manuscript and helped clarifying the conclusions. Interpretation of data has been adequately revised.

I am supportive of the publication of the manuscript at this stage, although writing needs improvement and simplification.

However, regarding the SASP analysis (new data included in revised Fig. 1f), it is intriguing that most cytokines are upregulated in 'young vs. old' HDFs. How do the authors reconcile these data with the transcriptome analysis showing increased proliferation and decreased SASP-related pathways?

In addition, I provide minor suggestions and comments to the authors that are annotated in the attached pdf file below.

REVIEWERS' COMMENTS:

Reviewer #1 (Remarks to the Author):

The authors have properly addressed all issues raised and therefore, I recommend publication.

[Response] We would like to thank you for your constructive comments on our manuscript.

Reviewer #2 (Remarks to the Author):

The authors have addressed all the major concerns raised by the Reviewers and the Editor. Substantial amount of new data have improved the quality of the manuscript and helped clarifying the conclusions. Interpretation of data has been adequately revised.

I am supportive of the publication of the manuscript at this stage, although writing needs improvement and simplification.

[Response] We would like to thank you for your constructive comments on our manuscript.

However, regarding the SASP analysis (new data included in revised Fig. 1f), it is intriguing that most cytokines are upregulated in 'young vs. old' HDFs. How do the authors reconcile these data with the transcriptome analysis showing increased proliferation and decreased SASP-related pathways?

[Response] We really appreciate for pointing out the discrepancy between the results from SASP and transcriptome analyses. Fig. 2c-d (see below) showed that “chemokine production” was enriched by 11 early down-regulated genes (E-DOWN), indicating that “chemokine production” was decreased by KU+Y. However, Fig. 1f (see below) showed that KU+Y increased SASPs, which is inconsistent with the decreased chemokine production in Fig. 2c-d.

To reconcile this discrepancy, we first checked which genes out of 11 E-DOWN genes had “chemokine production” GOBP annotations. Only two genes (POSTN and F2RL1) had “chemokine production” annotations, which suggests “weak enrichment” of chemokine production by the 11 E-DOWN genes, and they are not the known SASP genes or key genes involved in chemokine production. Second, we compared the SASP protein profile measured from the culture media using cytokine array with the mRNA profile of the cytokines on the cytokine array measured using microarray (see “comparison of secreted protein and mRNA profiles of cytokines” below). Many cytokines (e.g., PDGF BB and IGF-1) tended to show consistent up-regulation patterns by KU+Y treatment, although some of cytokines showed inconsistent protein and mRNA changes. These data suggest that KU+Y treatment tended to increase the levels of many cytokines.

Based on these data, we decided to change our criteria for “GOBPs significantly enriched by a gene list” to be GOBPs with enrichment $P < 0.05$ and the number of genes involved in the GOBPs ≥ 3 from enrichment $P < 0.05$ and the number of genes involved in the GOBPs ≥ 2 . When the new enrichment criteria were used, “chemokine production” was now removed from Fig. 2d and from the list of enriched GOBPs in Supplement Data S3. Accordingly, we removed “chemokine production” from the list of down-regulated SynAGs and “suggesting suppression of the SASPs” in the corresponding Results. Also, we revised the criteria in the corresponding Material and Methods.

Fig. 2c-d,

Fig. 1f

Comparison of secreted protein and mRNA profiles of cytokines

Revised Fig. 2d,

Results:

Transcriptome analysis reveals key candidate regulators of the synergistic effect of KU+Y

... These data collectively indicate that KU+Y leads to much more robust cell proliferation than KU or Y alone. On the other hand, the processes related to calcium ion homeostasis and protein transport (including extracellular vesicle and Golgi apparatus) were enriched with the downregulated SynAGs (E- and L-DOWN groups, respectively). ...

MATERIALS AND METHODS:

Functional enrichment analysis and TF enrichment analysis

... GO terms and KEGG pathways with a P value < 0.05 and count ≥ 3 were identified as enriched with the genes. ...

In addition, I provide minor suggestions and comments to the authors that are annotated in the attached pdf file below.

[Response] We accepted the reviewer's comments and revised the manuscript. In particular, we replaced all over the manuscript 'senescent HDFs' by 'senescent HDF cultures' or 'high passage HDFs'. For example, Results was revised as follows (underlined):

RESULTS

KU and Y synergistically restore the proliferation of senescent HDF cultures

We sought to examine the synergistic effects of KU and Y, which inhibit the kinase activity of ATM (autophosphorylation of ATM) and ROCK (phosphorylation of MYPT1), respectively, on the proliferation of senescent HDF cultures (Supplementary Fig. S1). To this end, we first determined the optimal concentrations of KU and Y for cotreatment as 0.5 and 2.5 μM , respectively. These concentrations resulted in the maximum proliferation of high passage HDFs at 15 days post treatment (DPT) among the four tested concentrations of each drug (0.125, 0.25, 0.5, and 1.5 μM for KU; 2.5, 5, 7.5, and 10 μM for Y) (Fig. 1a).